# Reinforcement Learning-Based Estimation for Partial Differential Equations

## Abstract

In systems governed by nonlinear partial differential equations such as fluid flows, the design of state estimators such as Kalman filters relies on a reduced-order model (ROM) that projects the original high-dimensional dynamics onto a computationally tractable low-dimensional space. However, ROMs are prone to large errors, which negatively affects the performance of the estimator. Here, we introduce the reinforcement learning reduced-order estimator (RL-ROE), a ROM-based estimator in which the correction term that takes in the measurements is given by a nonlinear policy trained through reinforcement learning. The nonlinearity of the policy enables the RL-ROE to compensate efficiently for errors of the ROM, while still taking advantage of the imperfect knowledge of the dynamics. Using examples involving the Burgers and Navier-Stokes equations, we show that in the limit of very few sensors, the trained RL-ROE outperforms a Kalman filter designed using the same ROM. Moreover, it yields accurate high-dimensional state estimates for reference trajectories corresponding to various physical parameter values, without direct knowledge of the latter.

## 1 Introduction

Active control of turbulent flows has the potential to cut down emissions across a range of industries through drag reduction in aircrafts and ships or improved efficiency of heating and air-conditioning systems, among many other examples (Brunton & Noack, 2015). But real-time feedback control requires inferring the state of the system from sparse measurements using an algorithm called a state estimator, which typically relies on a model for the underlying dynamics (Simon, 2006). Among state estimators, the Kalman filter is by far the most well-known thanks to its optimality for linear systems, which has led to its widespread use in numerous applications (Kalman, 1960; Zarchan, 2005). However, continuous systems such as fluid flows are governed by partial differential equations (PDEs) which, when discretized, yield high-dimensional and oftentimes nonlinear dynamical models with hundreds or thousands of state variables. These high-dimensional models are too expensive to integrate with common state estimation techniques, especially in the context of embedded systems. Thus, state estimators for control are instead designed based on a reduced-order model (ROM) of the system, in which the underlying dynamics are projected to a low-dimensional subspace that is computationally tractable (Barbagallo et al., 2009; Rowley & Dawson, 2017).

A big challenge is that ROMs provide a simplified and imperfect description of the dynamics, which negatively affects the performance of the state estimator. One potential solution is to improve the accuracy of the ROM through the inclusion of additional closure terms (Ahmed et al., 2021). In this paper, we leave the ROM untouched and instead propose a new design paradigm for the estimator itself, which we call a reinforcement-learning reduced-order estimator (RL-ROE). The RL-ROE is constructed from the ROM in an analogous way to a Kalman filter, with the crucial difference that the linear filter gain function, which takes in the current measurement data, is replaced by a nonlinear policy trained through reinforcement learning (RL). The flexibility of the nonlinear policy, parameterized by a neural network, enables the RL-ROE to compensate for errors of the ROM while still taking advantage of the imperfect knowledge of the dynamics. Indeed, we show that in the limit of sparse measurements, the trained RL-ROE outperforms a Kalman filter designed using the same ROM and displays robust estimation performance across different dynamical regimes. To our knowledge, the RL-ROE is the first application of RL to state estimation of parametric PDEs.

## 2 GENERAL METHODOLOGY

### 2.1 PROBLEM FORMULATION

Consider the parametric discrete-time nonlinear system given by

$$\boldsymbol{z}_{k+1} = \boldsymbol{f}(\boldsymbol{z}_k; \mu), \tag{1a}$$

$$\boldsymbol{y}_k = \boldsymbol{C}\boldsymbol{z}_k + \boldsymbol{n}_k, \tag{1b}$$

where $\boldsymbol{z}_k \in \mathbb{R}^n$ and $\boldsymbol{y}_k \in \mathbb{R}^p$ are respectively the state and measurement at time $k$, $\boldsymbol{f} : \mathbb{R}^n \to \mathbb{R}^n$ is a time-invariant nonlinear map from current to next state, $\boldsymbol{n}_k \in \mathbb{R}^p$ is observation noise (assumed zero unless stated otherwise), $\mu \in \mathbb{R}$ is a physical parameter, and $\boldsymbol{C} \in \mathbb{R}^{p \times n}$ is a linear map from state to measurement. In this study, we assume that the dynamics given in (1) are obtained from a high-fidelity numerical discretization of a nonlinear partial differential equation (PDE), which typically requires a large number $n$ of continuous state variables (on the order of at least a few hundreds). Nonetheless, our work is applicable to any high-dimensional nonlinear system of the form (1). We do not account for exogenous control inputs to the system, which is left for future work.

Here, we will focus on the post-transient dynamics of (1); these are the observed dynamics once the transients associated with the initial condition have died down. In particular, we consider systems whose post-transient dynamics are described by an attractor that is either a steady state, a periodic limit cycle or a quasi-periodic limit cycle, which encompasses the behavior of a large class of physical systems. The nature of the attractor is independent of the initial condition but depends on the value of $\mu$, which we will consider to be in a range $[\mu_1, \mu_2]$.

The purpose of the present work is to combine reduced-order modeling (ROM) and reinforcement learning (RL) to construct a state estimator that solves the following problem: given a sequence of measurements $\{\boldsymbol{y}_1, \cdots, \boldsymbol{y}_k\}$ from a post-transient reference trajectory of (1), estimate the high-dimensional state $\boldsymbol{z}_k$ at current time $k$ without knowledge of $\mu$ itself. The ROM procedure, which follows standard practices, is described in Section 2.2. The integration of the ROM with RL to solve the estimation problem, which constitutes the main novelty of the paper, is described in Section 2.3.

### 2.2 REDUCED-ORDER MODEL

Since the high dimensionality of (1) renders online estimation impractical, it is customary to formulate a reduced-order model (ROM) of the dynamics (Rowley & Dawson, 2017). First, one chooses a suitable linearly independent set of modes $\{\boldsymbol{u}_1, \ldots, \boldsymbol{u}_r\}$, where $\boldsymbol{u}_i \in \mathbb{R}^n$, defining an $r$-dimensional subspace of $\mathbb{R}^n$ in which most of the dynamics is assumed to take place. Stacking these modes as columns of a matrix $\boldsymbol{U} \in \mathbb{R}^{n \times r}$, one can then express $\boldsymbol{z}_k \simeq \boldsymbol{U}\boldsymbol{x}_k$, where the reduced-order state $\boldsymbol{x}_k \in \mathbb{R}^r$ represents the coordinates of $\boldsymbol{z}_k$ in the subspace. Finally, one finds a ROM for the dynamics of $\boldsymbol{x}_k$, which is vastly cheaper to evolve than (1) when $r \ll n$.

There exist various ways to find an appropriate set of modes $\boldsymbol{U}$ and corresponding ROM for the dynamics of $\boldsymbol{x}_k$ (Taira et al., 2017). In this work, we employ the Dynamic Mode Decomposition (DMD), a purely data-driven algorithm that has found numerous applications in fields ranging from fluid dynamics to neuroscience (Schmid, 2010; Kutz et al., 2016). Importantly, we seek a single ROM to describe dynamics corresponding to various parameter values $\mu \in [\mu_1, \mu_2]$ since the state estimator that we will later construct based on this ROM does not have knowledge of $\mu$.

In order to apply DMD, we first construct a training dataset by solving (1) for values of $\mu$ belonging to a finite set $S \subset [\mu_1, \mu_2]$, resulting in a concatenated collection of snapshots $\boldsymbol{Z}_{\text{train}} = \{\boldsymbol{Z}^\mu\}_{\mu \in S}$, where each $\boldsymbol{Z}^\mu = \{\boldsymbol{z}_0^\mu, \ldots, \boldsymbol{z}_K^\mu\}$ is a post-transient trajectory of (1a) for a specific value $\mu \in S$. The DMD then seeks a best-fit linear model of the dynamics in the form of a matrix $\boldsymbol{A} \in \mathbb{R}^{n \times n}$ such that $\boldsymbol{z}_{k+1}^\mu \simeq \boldsymbol{A}\boldsymbol{z}_k^\mu$ for all $k$ and $\mu$, and computes the modes $\boldsymbol{U}$ as the $r$ leading principal component analysis (PCA) modes of $\boldsymbol{Z}_{\text{train}}$. The transformation $\boldsymbol{z}_k \simeq \boldsymbol{U}\boldsymbol{x}_k$ and the orthogonality of $\boldsymbol{U}$ then yield a linear discrete-time ROM of the form

$$\boldsymbol{x}_k = \boldsymbol{A}_r \boldsymbol{x}_{k-1} + \boldsymbol{w}_{k-1}, \tag{2a}$$

$$\boldsymbol{y}_k = \boldsymbol{C}_r \boldsymbol{x}_k + \boldsymbol{v}_k, \tag{2b}$$

where $\boldsymbol{A}_r = \boldsymbol{U}^\mathsf{T}\boldsymbol{A}\boldsymbol{U} \in \mathbb{R}^{r \times r}$ and $\boldsymbol{C}_r = \boldsymbol{C}\boldsymbol{U} \in \mathbb{R}^{p \times r}$ are the reduced-order state-transition and observation models, respectively. The (unknown) non-Gaussian process noise $\boldsymbol{w}_k$ and observation

noise $\boldsymbol{v}_k$ account for the neglected PCA modes of $\boldsymbol{Z}_{\text{train}}$ in $\boldsymbol{U}$, as well as the error incurred by the linear approximation and effective averaging of the dynamics over a range of $\mu$. Additional details regarding the calculation of $\boldsymbol{A}_r$ and $\boldsymbol{U}$ are provided in Appendix A of the supplementary materials.

## 2.3 REINFORCEMENT LEARNING-BASED REDUCED-ORDER ESTIMATOR

Using the ROM (2) defined by $\boldsymbol{A}_r$, $\boldsymbol{C}_r$ and $\boldsymbol{U}$, we now want to solve the estimation problem defined in Section 2.1. To this effect, we design a reduced-order estimator (ROE) of the form

$$\hat{\boldsymbol{x}}_k = \boldsymbol{A}_r \hat{\boldsymbol{x}}_{k-1} + \boldsymbol{a}_k, \tag{3a}$$

$$\boldsymbol{a}_k \sim \boldsymbol{\pi}_{\boldsymbol{\theta}}(\,\cdot\,|\boldsymbol{y}_k, \hat{\boldsymbol{x}}_{k-1}), \tag{3b}$$

where $\hat{\boldsymbol{x}}_k$ is an estimate of the reduced-order state $\boldsymbol{x}_k$ and $\boldsymbol{a}_k \in \mathbb{R}^r$ is an action sampled from a nonlinear and stochastic policy $\boldsymbol{\pi}_{\boldsymbol{\theta}}$, which takes as input the current measurement $\boldsymbol{y}_k$ from the reference trajectory of (1a) and the previous state estimate $\hat{\boldsymbol{x}}_{k-1}$. The subscript $\boldsymbol{\theta}$ denotes the set of parameters that define the policy, whose goal is to use the sparse measurements $\boldsymbol{y}_k$ to act on the dynamics of $\hat{\boldsymbol{x}}_k$ in (3a) so that the reconstructed high-dimensional state estimate $\hat{\boldsymbol{z}}_k = \boldsymbol{U}\hat{\boldsymbol{x}}_k$ converges towards the (hidden) reference state $\boldsymbol{z}_k$. Note that designing state estimators, also called state observers, by correcting the dynamics model with a measurement-dependent term is a standard approach in control theory (Korovin & Fomichev, 2009; Besançon, 2007).

A Kalman filter is a special case of such an estimator, for which the action in (3b) is given by

$$\boldsymbol{a}_k = \boldsymbol{K}_k(\boldsymbol{y}_k - \boldsymbol{C}_r \boldsymbol{A}_r \hat{\boldsymbol{x}}_{k-1}), \tag{4}$$

with $\boldsymbol{K}_k \in \mathbb{R}^{r \times p}$ the optimal Kalman gain. Although the Kalman filter is the optimal linear filter for linear systems (Julier & Uhlmann, 2004; Simon, 2006), its performance suffers in the presence of unmodeled dynamics and parameter uncertainty, both of which are present in our case. Thus, this motivates the adoption of the more general form (3b), which retains the dependence of $\boldsymbol{a}_k$ on $\boldsymbol{y}_k$ and $\hat{\boldsymbol{x}}_{k-1}$ but is more flexible thanks to the nonlinearity of the policy $\boldsymbol{\pi}_{\boldsymbol{\theta}}$.

We first train the policy $\boldsymbol{\pi}_{\boldsymbol{\theta}}$ in an offline phase, using deep RL to solve the optimization problem

$$\boldsymbol{\theta}^* = \arg\min_{\boldsymbol{\theta}} \mathbb{E}\left[\sum_{k=1}^{K} ||\boldsymbol{z}_k - \boldsymbol{U}\hat{\boldsymbol{x}}_k||^2 + \lambda||\boldsymbol{a}_k||^2\right], \tag{5}$$

where the expectation is taken over initial estimates $\hat{\boldsymbol{x}}_0$, initial true states $\boldsymbol{z}_0$, parameters $\mu$, estimate trajectories $\{\hat{\boldsymbol{x}}_1, \hat{\boldsymbol{x}}_2, \dots\}$ induced by $\boldsymbol{\pi}_{\boldsymbol{\theta}}$ through (3), and true trajectories of states $\{\boldsymbol{z}_1, \boldsymbol{z}_2, \dots\}$ and measurements $\{\boldsymbol{y}_1, \boldsymbol{y}_2, \dots\}$ induced by (1). The first squared term in (5) penalizes the error between the high-dimensional estimate $\hat{\boldsymbol{z}}_k = \boldsymbol{U}\hat{\boldsymbol{x}}_k$ and the true $\boldsymbol{z}_k$. The second squared term favors smaller values for the action $\boldsymbol{a}_k$, which acts as a regularization mechanism. Unless indicated otherwise, we will consider $\lambda = 0$. By considering different values of $\mu$ during training, a strategy called domain randomization (Peng et al., 2018b), we ensure robustness of the policy with respect to $\mu$ during online deployment of the estimator. Note that the stochasticity of $\boldsymbol{\pi}_{\boldsymbol{\theta}}$ lets the RL algorithm explore different actions during the training process, but is turned off during online deployment.

We call the estimator constructed and trained through this process an RL-trained ROE, or RL-ROE for short. Finally, an interpretation of the estimator dynamics (3) and the optimization problem (5) in the context of Bayesian inference is presented in Appendix B.

## 2.4 SUMMARY OF THE PROPOSED METHODOLOGY

In summary, the methodology we propose consists of the following three steps. The first two are carried out offline using a training dataset $\boldsymbol{Z}_{\text{train}} = \{\boldsymbol{Z}^{\mu}\}_{\mu \in S}$ containing high-dimensional state snapshots from multiple trajectories of (1) for various $\mu$. The third takes place online using sole knowledge of measurements $\{\boldsymbol{y}_1, \dots, \boldsymbol{y}_k\}$ from a trajectory of (1) corresponding to unknown $\mu$ not necessarily belonging to $S$.

1. **Construction of the ROM (offline)**. A ROM of the form (2) is obtained by applying the DMD to the training dataset $\boldsymbol{Z}_{\text{train}}$.
2. **Training of the RL-ROE (offline)**. An RL-ROE of the form (3) is designed based upon the ROM constructed in Step 1, and its policy $\boldsymbol{\pi}_{\boldsymbol{\theta}}$ is trained using the reference trajectories contained in $\boldsymbol{Z}_{\text{train}}$.

3. **Deployment of the RL-ROE (online)**. Using measurements $\{\boldsymbol{y}_1, \ldots, \boldsymbol{y}_k\}$ from a reference trajectory of (1) corresponding to unknown $\mu$, the trained RL-ROE returns an estimate $\hat{\boldsymbol{z}}_k = \boldsymbol{U}\hat{\boldsymbol{x}}_k$ of the (unobserved) high-dimensional state $\boldsymbol{z}_k$.

The RL-ROE, which combines a ROM with a nonlinear policy $\boldsymbol{\pi_\theta}$ trained in an RL framework to solve the estimation problem, constitutes the main contribution of the present paper. The training procedure for $\boldsymbol{\pi_\theta}$, which involves a non-trivial reformulation of the time-varying estimation problem into a stationary Markov decision process, is described in the next section.

## 3 OFFLINE TRAINING METHODOLOGY

In order to train $\boldsymbol{\pi_\theta}$ with reinforcement learning, we need to formulate the optimization problem (5) as a stationary Markov decision process (MDP). However, this is no trivial task given that the aim of the policy is to minimize the error between the state estimate $\hat{\boldsymbol{z}}_k = \boldsymbol{U}\hat{\boldsymbol{x}}_k$ and a time-dependent reference state $\boldsymbol{z}_k$. At first sight, such trajectory tracking problem requires a time-dependent reward function and, therefore, a non-stationary MDP (Puterman, 2014; Lecarpentier & Rachelson, 2019). To be able to use off-the-shelf RL algorithms, we introduce a trick to translate this non-stationary MDP to an equivalent stationary MDP based on an extended state. Specifically, we show hereafter that the problem can be framed as a stationary MDP whose state has been enhanced with $\boldsymbol{z}_k$ and $\mu$, removing the time dependence from the reward function.

Letting $\boldsymbol{s}_k = (\hat{\boldsymbol{x}}_{k-1}, \boldsymbol{z}_k, \mu) \in \mathbb{R}^{r+n+1}$ denote an augmented state at time $k$, we can define an MDP consisting of the tuple $(\mathcal{S}, \mathcal{A}, \mathcal{P}, \mathcal{R})$, where $\mathcal{S} = \mathbb{R}^{r+n+1}$ is the augmented state space, $\mathcal{A} \subset \mathbb{R}^r$ is the action space, $\mathcal{P}(\cdot|\boldsymbol{s}_k, \boldsymbol{a}_k)$ is a transition probability, and $\mathcal{R}(\boldsymbol{s}_k, \boldsymbol{a}_k, \boldsymbol{s}_{k+1})$ is a reward function. At each time step $k$, the agent selects an action $\boldsymbol{a}_k \in \mathcal{A}$ according to the policy $\boldsymbol{\pi_\theta}$ defined in (3b), which can be expressed as

$$\boldsymbol{a}_k \sim \boldsymbol{\pi_\theta}(\,\cdot\,|\boldsymbol{o}_k), \tag{6}$$

where $\boldsymbol{o}_k = (\boldsymbol{y}_k, \hat{\boldsymbol{x}}_{k-1}) = (\boldsymbol{C}\boldsymbol{z}_k, \hat{\boldsymbol{x}}_{k-1})$ is a partial observation of the current state $\boldsymbol{s}_k$. The state $\boldsymbol{s}_{k+1} = (\hat{\boldsymbol{x}}_k, \boldsymbol{z}_{k+1}, \mu)$ at the next time step is then obtained from equations (1a) and (3a) as

$$\boldsymbol{s}_{k+1} = (\boldsymbol{A}_r \hat{\boldsymbol{x}}_{k-1} + \boldsymbol{a}_k, \boldsymbol{f}(\boldsymbol{z}_k; \mu), \mu), \tag{7}$$

which defines the transition model $\boldsymbol{s}_{k+1} \sim \mathcal{P}(\cdot|\boldsymbol{s}_k, \boldsymbol{a}_k)$. Finally, the agent receives the reward

$$r_k = \mathcal{R}(\boldsymbol{s}_k, \boldsymbol{a}_k, \boldsymbol{s}_{k+1}) = -||\boldsymbol{z}_k - \boldsymbol{U}\hat{\boldsymbol{x}}_k||^2 - \lambda ||\boldsymbol{a}_k||^2, \tag{8}$$

which is minus the term to be minimized at each step in (5). Thanks to the incorporation of $\boldsymbol{z}_k$ into $\boldsymbol{s}_k$, the reward function (8) has no explicit time dependence and the MDP is therefore stationary.

The RL training process then finds the optimal policy parameters

$$\boldsymbol{\theta}^* = \arg\max_{\boldsymbol{\theta}} \,\mathbb{E}_{\tau \sim \boldsymbol{\pi_\theta}} [R(\tau)], \tag{9}$$

where the expectation is over trajectories $\tau = (\boldsymbol{s}_1, \boldsymbol{a}_1, \boldsymbol{s}_2, \boldsymbol{a}_2, \ldots)$, and $R(\tau) = \sum_{k=1}^{K} r_k$ is the finite-horizon undiscounted return. Thus, the optimization problem (9) solved by RL is equivalent to that stated in (5). At the beginning of every episode, the environment is reset according to the distributions

$$\hat{\boldsymbol{x}}_0 \sim \boldsymbol{p}_{\hat{\boldsymbol{x}}_0}(\cdot), \quad \boldsymbol{z}_0 \sim \boldsymbol{p}_{\boldsymbol{z}_0}(\cdot), \quad \mu \sim p_\mu(\cdot), \tag{10}$$

from which the augmented state $\boldsymbol{s}_1 = (\hat{\boldsymbol{x}}_0, \boldsymbol{z}_1, \mu) = (\hat{\boldsymbol{x}}_0, \boldsymbol{f}(\boldsymbol{z}_0; \mu), \mu)$ follows immediately; $\boldsymbol{s}_1$ thus constitutes the start of the agent-environment interactions. The distribution $\boldsymbol{p}_{\hat{\boldsymbol{x}}_0}(\cdot)$ bestows robustness of the learned policy with respect to the initial state estimate $\hat{\boldsymbol{x}}_0$, while the distributions $\boldsymbol{p}_{\boldsymbol{z}_0}(\cdot)$ and $p_\mu(\cdot)$ enable the same policy $\boldsymbol{\pi_\theta}$ to be trained on several reference trajectories of (1) corresponding to various $\mu \in [\mu_1, \mu_2]$. In practice, we reuse the same reference trajectories corresponding to $\mu \in S$ from the training dataset $\boldsymbol{Z}$ utilized to construct the DMD in Section 2.2, so that we do not have to keep solving (1a) during the training process. Thus, at the beginning of every episode, we draw a random $\mu$ from $S$ and we initialize $\boldsymbol{z}_0$ as the starting state $\boldsymbol{z}_0^\mu$ of the corresponding trajectory $\boldsymbol{Z}^\mu = \{\boldsymbol{z}_0^\mu, \ldots, \boldsymbol{z}_K^\mu\}$.

To learn $\boldsymbol{\theta}^*$, we employ the Proximal Policy Optimization (PPO) algorithm (Schulman et al., 2017), which belongs to the class of policy gradient methods (Sutton et al., 2000). The parameterization of the policy $\boldsymbol{\pi_\theta}$, implementation details and training hyperparameters are discussed in Appendix D.

*Remark.* Since the policy (6) is conditioned on a partial observation $\boldsymbol{o}_k$ of the state $\boldsymbol{s}_k$, the stationary MDP we have defined in this section is, in fact, a partially observable MDP (POMDP). In this case, it is known that the globally optimal policy depends on a summary of the history of past observations and actions, $\boldsymbol{h}_k = \{\boldsymbol{o}_1, \boldsymbol{a}_1, \ldots, \boldsymbol{o}_k\}$, rather than just the current observation $\boldsymbol{o}_k$ (Kaelbling et al., 1998). However, policies formulated based on an incomplete summary of $\boldsymbol{h}_k$ are common in practice and still achieve good results (Sutton & Barto, 2018). We therefore pursue this approach in the present paper, and leave for future work testing the generalization of our policy input to a more complete summary of $\boldsymbol{h}_k$. We also note that policy gradient methods, which PPO belongs to, do not require the Markov property of the state (that is, conditional independence of future states on past states given the present state) and can therefore be readily applied to the POMDP setting. For our problem, this guarantees that the PPO algorithm will converge to a locally optimum policy.

## 4 RESULTS

We evaluate the state estimation performance of the RL-ROE for systems governed by the Burgers equation and Navier-Stokes equations. For each system, we first compute various solution trajectories corresponding to different physical parameter values, which we use to construct the ROM and train the RL-ROE following the procedure outlined in Section 2.4. The trained RL-ROE is finally deployed online and compared against a time-dependent Kalman filter constructed from the same ROM, which we refer to as KF-ROE. The KF-ROE is given by equations (3a) and (4), with the calculation of the time-varying Kalman gain detailed in Appendix C of the supplementary materials.

Before proceeding to the results, we discuss our choice of baseline. The ensemble Kalman filter and 4D-Var are two estimation techniques for high-dimensional systems such as those governed by PDEs (Lorenc, 2003). Although they are commonly employed for data assimilation in numerical weather prediction, they require large computational resources since they involve repeated solutions of the high-dimensional dynamics (1). Thus, they are not applicable in the context of embedded control systems, whose limited resources call for an inexpensive model such as the ROM (2). Since the ROM that we consider has linear dynamics, extensions of the Kalman filter for nonlinear dynamics such as the extended or unscented Kalman filters (Wan & Van Der Merwe, 2000; Julier & Uhlmann, 2004) are not relevant, and the vanilla Kalman filter remains the best choice of baseline.

### 4.1 BURGERS EQUATION

The forced Burgers equation is a prototypical nonlinear hyperbolic PDE that takes the form

$$\frac{\partial u}{\partial t} + u\frac{\partial u}{\partial x} - \nu\frac{\partial^2 u}{\partial x^2} = f(x, t), \tag{11}$$

where $u(x, t)$ is the velocity at position $x \in [0, L]$ and time $t$, $f(x, t)$ is a distributed time-dependent forcing, and the scalar $\nu$ acts like a viscosity. Here, we choose a forcing of the form

$$f(x, t) = 2\sin(\omega t - kx) + 2\sin(3\omega t - kx) + 2\sin(5\omega t - kx), \tag{12}$$

where $k = 2\pi/L$, and we let $\nu$ and $\omega$ be related through a scalar parameter $\mu \in [0, 1]$ as follows:

$$\nu = \nu_1 + (\nu_2 - \nu_1)\mu, \qquad \omega = \omega_1 + (\omega_2 - \omega_1)\mu. \tag{13a}$$

Thus, $\mu$ can be regarded as a physical parameter that affects the dynamics of the forced Burgers equation through both $\nu$ and $\omega$. We consider periodic boundary conditions and choose $L = 1$, $\nu_1 = 0.01$, $\nu_2 = 0.1$, $\omega_1 = 0.2\pi$, $\omega_1 = 0.4\pi$.

We solve the forced Burgers equation using a spectral method with $n = 256$ Fourier modes and a fifth-order Runge-Kutta time integration scheme. We define the discrete-time state vector $\boldsymbol{z}_k \in \mathbb{R}^n$ that contains the values of $u$ at $n$ equally-spaced collocation points and at discrete time steps $t = k\Delta t$, where $\Delta t = 0.05$. To generate the training dataset $\boldsymbol{Z}_{\text{train}} = \{\boldsymbol{Z}^\mu\}_{\mu \in S}$ used for constructing the ROM and training the RL-ROE, we compute solutions of the Burgers equation corresponding to $\mu \in S = \{0, 0.1, 0.2, \ldots, 0.1\}$. For each $\mu$, we discard the transient portion of the dynamics and save 201 snapshots $\boldsymbol{Z}^\mu = \{\boldsymbol{z}_0^\mu, \ldots, \boldsymbol{z}_{200}^\mu\}$ in the post-transient regime. We retain $r = 10$ modes when constructing the ROM, corresponding to an-order-of-magnitude reduction in the dimensionality of the system. We train the RL-ROE using episodes of length $K = 200$ steps to make full use of the trajectories stored in $\boldsymbol{Z}_{\text{train}}$, and we end the training process when the return no longer increases

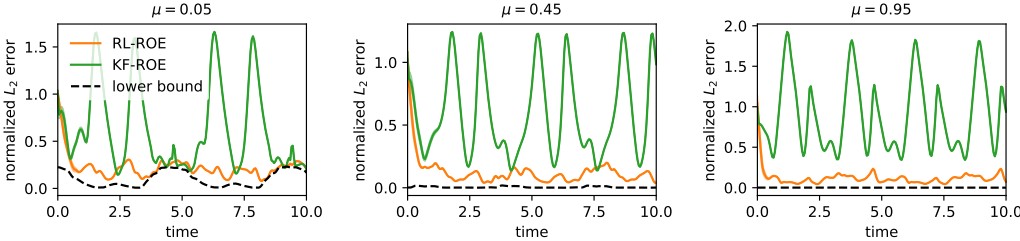

Figure 1: Burgers equation with $p = 4$ sensors. Normalized $L_2$ error of the RL-ROE and KF-ROE for the estimation of trajectories corresponding to values of $\mu$ not seen during training.

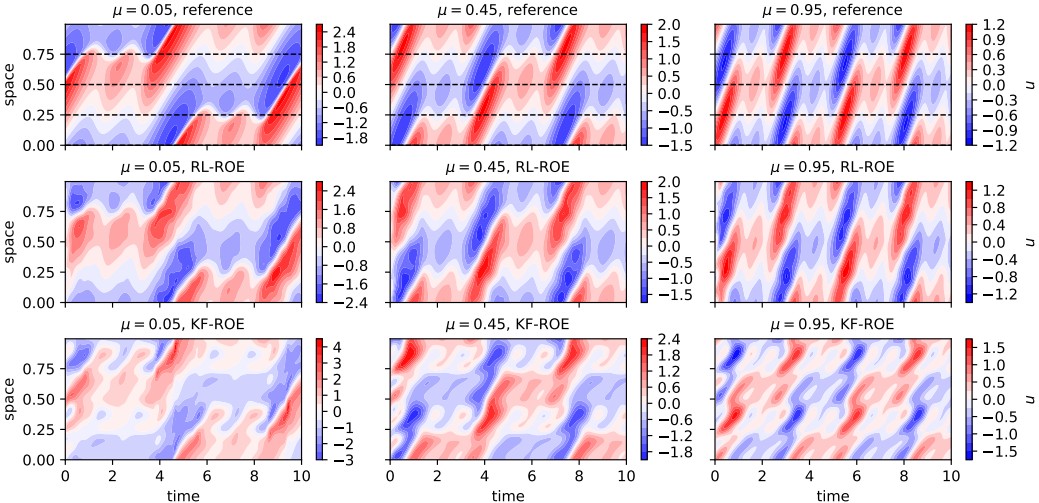

Figure 2: Burgers equation with $p = 4$ sensors. Reference trajectories for values of $\mu$ not seen during training and corresponding RL-ROE and KF-ROE estimates. The dashed lines on the reference trajectory plots indicate the sensor data seen by the RL-ROE and KF-ROE.

on average. The RL hyperparameters and learning curve displaying the performance improvement of the RL-ROE during the training process are reported in Appendix D.

The trained RL-ROE and the KF-ROE are now compared based on their ability to track reference trajectories corresponding to various $\mu$ using sparse measurements of $u$ from a limited number $p$ of equally-spaced sensors (Appendix E describes the corresponding $C$). The evaluation is carried out using 20 initial state estimates sampled from a Gaussian distribution with unit standard deviation.

Beginning with $p = 4$ sensors, Figure 1 reports the mean (lines) and standard deviation (shaded areas) of the normalized $L_2$ error for 3 values of $\mu$ not present in the training dataset $\boldsymbol{Z}_{\mathrm{train}}$ used to construct the ROM and train the RL-ROE. The normalized $L_2$ error is defined as $||\hat{\boldsymbol{z}}_k - \boldsymbol{z}_k||/||\boldsymbol{z}_k||$, where $\boldsymbol{z}_k$ is the high-dimensional reference solution and $\hat{\boldsymbol{z}}_k = \boldsymbol{U}\hat{\boldsymbol{x}}_k$ is the high-dimensional reconstruction of the RL-ROE or KF-ROE estimate $\hat{\boldsymbol{x}}_k$. The error of the RL-ROE is very close to the lower bound, which is the error incurred by projecting the reference solution $\boldsymbol{z}_k$ to the modes $\boldsymbol{U}$, i.e. the lowest possible error achievable by an ROE based on $\boldsymbol{U}$. Spatio-temporal contours of the reference solutions for the same 3 values of $\mu$ and the corresponding RL-ROE and KF-ROE high-dimensional estimates are shown in Figure 2. The RL-

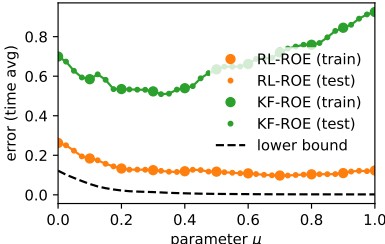

Figure 3: Burgers equation with $p = 4$ sensors. Time average of the normalized $L_2$ error versus $\mu$. Values of $\mu$ present in $\boldsymbol{Z}_{\mathrm{train}}$ shown by large circles.

ROE vastly outperforms the KF-ROE, which demonstrates the superiority of a nonlinear correction to the estimator dynamics (3a). We emphasize that the Kalman filter was tuned to obtain the best possible results for the KF-ROE (see Appendix C for details on the tuning process).

Figure 3 reports the time average of the normalized $L_2$ error as a function of $\mu$ for $p = 4$, with the values present in $\boldsymbol{Z}_{\text{train}}$ indicated by large circles. The RL-ROE exhibits robust performance across the entire parameter range $\mu \in [0, 1]$, including when estimating previously unseen trajectories.

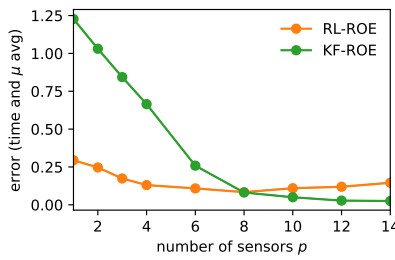

Figure 4: Burgers equation. Average over time and over $\mu$ of the normalized $L_2$ error versus number $p$ of sensors.

Finally, Figure 4 displays the average over time and over $\mu$ of the normalized $L_2$ error for varying number $p$ of sensors. Note that each value of $p$ corresponds to a separately trained RL-ROE. As the number of sensors increases, the KF-ROE performs better and better until its accuracy overtakes that of the RL-ROE. We hypothesize that the accuracy of the RL-ROE is limited by the inability of the RL training process to find an optimal policy, due to both the non-convexity of the optimization landscape as well as shortcomings inherent to current deep RL algorithms. This being said, the strength of the nonlinear policy of the RL-ROE becomes very clear in the very sparse sensing regime; its performance remains remarkably robust as the number of sensors reduces to 2 or even 1. In Appendix F, spatio-temporal contours (similar as in Figure 2) of the reference solution and corresponding estimates for $p = 2$ and 12 illustrate that the slight advantage held by the KF-ROE for $p = 12$ is reversed into clear superiority of the RL-ROE for $p = 4$.

## 4.2 NAVIER-STOKES EQUATIONS

The Navier-Stokes equations are a set of nonlinear PDEs that describe the motion of fluids flows. For incompressible fluids, the Navier-Stokes equations take the form

$$\frac{\partial \boldsymbol{u}}{\partial t} + (\boldsymbol{u} \cdot \nabla)\boldsymbol{u} = -\nabla p + \frac{1}{Re}\Delta\boldsymbol{u}, \tag{14a}$$

$$\nabla \cdot \boldsymbol{u} = 0, \tag{14b}$$

where $\boldsymbol{u}(\boldsymbol{x}, t)$ and $p(\boldsymbol{x}, t)$ are the velocity vector and pressure at position $\boldsymbol{x}$ and time $t$, and the scalar $Re$ is the Reynolds number. We consider the classical problem of a flow past a cylinder in a 2D domain, which is well known to exhibit a Hopf bifurcation from a steady wake to periodic vortex shedding at a critical Reynolds number $Re_c \sim 40$ (Jackson, 1987). For our study, we focus on the range $Re \in [10, 110]$, which makes the estimation problem very challenging since this range includes the bifurcation and therefore comprises solution trajectories with very different dynamics – steady for $Re < Re_c$, periodic limit cycle for $Re > Re_c$. Furthermore, the shedding frequency and spacing between consecutive vortices in the limit cycle regime both vary with $Re$.

We solve the Navier-Stokes equations with the open source finite volume code OpenFOAM using a mesh consisting of 18840 nodes and a second-order implicit scheme with time step 0.05. The discrete-time state vector $\boldsymbol{z}_k \in \mathbb{R}^{37680}$ contains the two velocity components of $\boldsymbol{u}$ at discrete time steps $t = k\Delta t$, where we choose $\Delta t = 0.25$. To generate the training dataset $\boldsymbol{Z}_{\text{train}} = \{\boldsymbol{Z}^{Re}\}_{Re \in S}$ for constructing the ROM and training the RL-ROE, we run simulations of the Navier-Stokes equations for $Re \in S = \{10, 20, 30, \ldots, 110\}$. We discard the transient portion of the dynamics (for the cases $Re > Re_c$) and save 201 snapshots $\boldsymbol{Z}^{Re} = \{\boldsymbol{z}_0^{Re}, \ldots, \boldsymbol{z}_{200}^{Re}\}$ in the post-transient regime. We retain $r = 20$ modes when constructing the ROM, corresponding to a three-orders-of-magnitude reduction in the dimensionality of the system. These modes are shown in Appendix G. We train the RL-ROE using episodes of length $K = 200$ steps to make full use of the trajectories stored in $\boldsymbol{Z}_{\text{train}}$ and end the training process when the return no longer increases on average. The RL hyperparameters and learning curve displaying the performance improvement of the RL-ROE during the training process are reported in Appendix D.

The trained RL-ROE and the KF-ROE are now compared based on their ability to track reference trajectories corresponding to various $Re$ using sparse measurements of $\boldsymbol{u}$ from a limited number $p$ of sensors randomly distributed in the wake region behind the cylinder (see Appendix E for the construction of the corresponding $\boldsymbol{C}$). The evaluation is carried out using 5

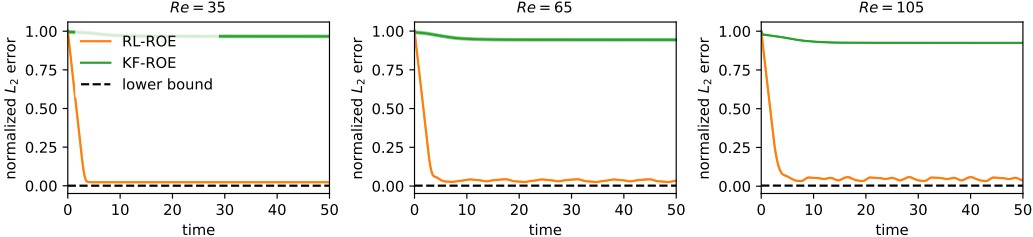

Figure 5: Navier-Stokes equations with $p = 3$ sensors. Normalized $L_2$ error of the RL-ROE and KF-ROE for the estimation of trajectories corresponding to values of $Re$ not seen during training.

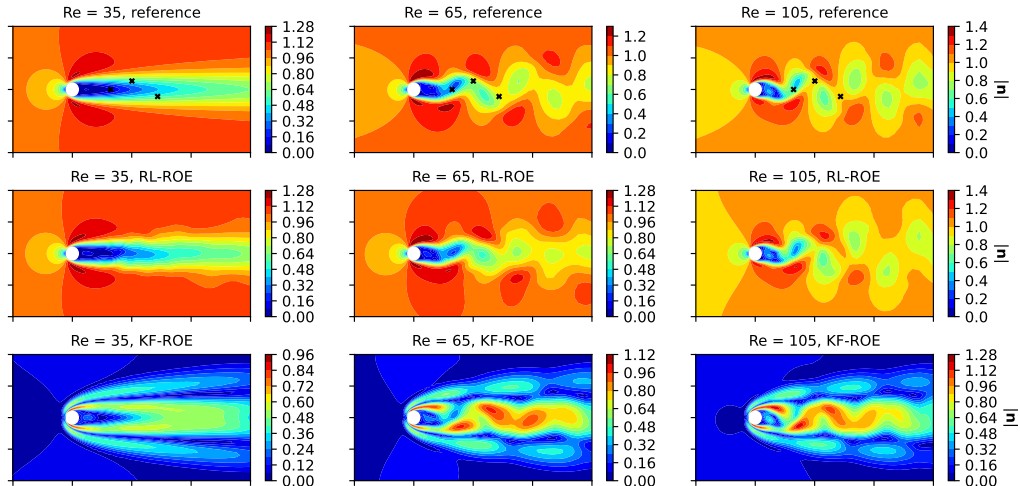

Figure 6: Navier-Stokes equations with $p = 3$ sensors. Velocity magnitude at $t = 50$ of the reference trajectories for values of $Re$ not seen during training and corresponding RL-ROE and KF-ROE estimates. The black crosses in the contours of the reference solutions indicate the sensor locations.

initial state estimates sampled from a Gaussian distribution with unit standard deviation. Beginning with $p = 3$ sensors, Figure 1 reports the mean (lines) and standard deviation (shaded areas) of the normalized $L_2$ error for 3 values of $Re$ not present in the training dataset $\mathbf{Z}_{\text{train}}$ used to construct the ROM and train the RL-ROE. The RL-ROE is almost as accurate as the lower bound, while the KF-ROE has normalized error around 1, which indicates complete failure.

The velocity magnitude of the reference solutions for the same 3 values of $Re$ and the corresponding RL-ROE and KF-ROE reconstructed high-dimensional estimates are shown at $t = 50$ in Figure 6. Remarkably, the RL-ROE is manages to estimate very precisely the entire flow field across different dynamical regimes, with the steady wake at $Re = 35$ being reproduced equally well as the wake of vortices at $Re = 65$ and 105. The KF-ROE, on the other hand, struggles to estimate the flow fields for all 3 Reynolds numbers, and instead predicts a velocity field that is almost everywhere zero except in the wake. Again, the superiority of the RL-ROE is granted by the nonlinearity of its policy – in fact, note that bifurcations such as the one exhibited by this flow are inherently nonlinear phenomena. Appendix H shows corresponding results in the presence of non-zero observation noise.

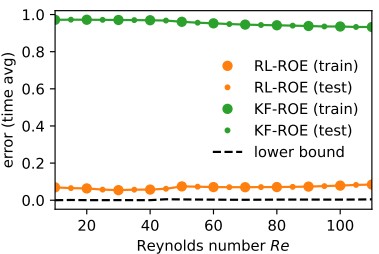

Figure 7: Navier-Stokes equations with $p = 3$ sensors. Time average of the normalized $L_2$ error versus $Re$. Values of $Re$ present in $\mathbf{Z}_{\text{train}}$ shown by large circles.

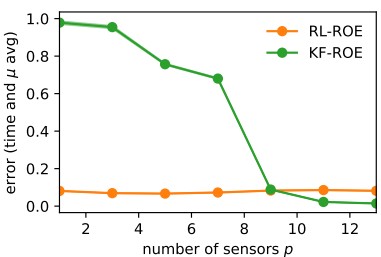

Figure 8: Navier-Stokes equations. Average over time and over $Re$ of the normalized $L_2$ error versus number $p$ of sensors.

Figure 7 reports the time average of the normalized $L_2$ error as a function of $\mu$ for $p = 3$, with the values present in $\boldsymbol{Z}_{\text{train}}$ indicated by large circles. The RL-ROE exhibits robust performance across the entire range of Reynolds numbers, including in the vicinity of the bifurcation at $Re_c \sim 40$ and for values of $Re$ not seen during training.

Figure 8 displays the average over time and over $Re$ of the normalized $L_2$ error for varying number $p$ of sensors. Although the KF-ROE eventually becomes very accurate in the presence of a large number $p$ of sensors, the accuracy of the RL-ROE remains remarkably stable as $p$ decreases, allowing it to vastly outperforms the KF-ROE as soon as $p < 8$. Once again, this showcases the benefits of using a nonlinear policy to correct the estimator dynamics (3a).

## 5 RELATED WORK

Previous studies have already proposed designing state estimators using policies trained through reinforcement learning. Morimoto & Doya (2007) introduced an estimator of the form $\hat{\boldsymbol{x}}_k = \boldsymbol{f}(\hat{\boldsymbol{x}}_{k-1}) + \boldsymbol{L}(\hat{\boldsymbol{x}}_{k-1})(\boldsymbol{y}_{k-1} - \boldsymbol{C}\hat{\boldsymbol{x}}_{k-1})$, where $\boldsymbol{f}(\cdot)$ is the state-transition model of the system, and the state-dependent filter gain matrix $\boldsymbol{L}(\hat{\boldsymbol{x}}_{k-1})$ is defined using Gaussian basis functions whose parameters are learned through a variant of vanilla policy gradient. Hu et al. (2020) proposed an estimator of the form $\hat{\boldsymbol{x}}_k = \boldsymbol{f}(\hat{\boldsymbol{x}}_{k-1}) + \boldsymbol{L}(\boldsymbol{x}_k - \hat{\boldsymbol{x}}_k)(\boldsymbol{y}_k - \boldsymbol{C}\boldsymbol{f}(\hat{\boldsymbol{x}}_{k-1}))$, where $\boldsymbol{L}(\boldsymbol{x}_k - \hat{\boldsymbol{x}}_k)$ is approximated by neural networks trained with a modified Soft-Actor Critic algorithm (Haarnoja et al., 2018). Although they derived convergence properties for the estimate error, the dependence of the filter gain $\boldsymbol{L}(\boldsymbol{x}_k - \hat{\boldsymbol{x}}_k)$ on the reference state $\boldsymbol{x}_k$ limits its practical application. Furthermore, a major difference between these past studies and our work is that they only consider low-dimensional systems with four state variables at most. Our RL-ROE, on the other hand, handles parametric PDEs described by tens of thousands of state variables as shown in the previous section.

In another line of works, reinforcement learning has been applied to learn control policies for joint torques that enable simulated characters to imitate given reference motions consisting of a sequence of target poses; see e.g. Peng et al. (2018a) or Lee et al. (2019). These are essentially trajectory tracking problems, as the policy learns to drive the character's pose towards that defined by the reference motion. Similar to us, these studies also propose to transform the problem into a stationary MDP by augmenting the state; but they do so by appending a scalar phase variable $\phi \in [0, 1]$ that represents the normalized time elapsed in the reference motion. The reward can then be formulated in terms of $q(\phi)$, where $q(\cdot)$ describes the reference motion, and no explicit time dependence appears. However, this approach would not work for our purpose since we do not wish to restrict the RL-ROE to estimating a single reference trajectory for each value of $\mu$. Many dynamical systems indeed admit multiple post-transient trajectories for given parameter values (Cross & Hohenberg, 1993). Augmenting the MDP's state with the entire snapshot from the reference trajectory instead of just time ensures that the policy $\boldsymbol{\pi_\theta}$ can learn any number of reference trajectories for each $\mu$.

## 6 CONCLUSIONS

In this paper, we have introduced the reinforcement learning reduced-order estimator (RL-ROE), a new state estimation methodology for parametric PDEs. Our approach relies on the construction of a computationally inexpensive reduced-order model (ROM) to approximate the dynamics of the system. The novelty of our contribution lies in the design, based on this ROM, of a reduced-order estimator (ROE) in which the filter correction term is given by a nonlinear stochastic policy trained offline through reinforcement learning. We demonstrate using simulations of the Burgers and Navier-Stokes equations that in the limit of very few sensors, the trained RL-ROE vastly outperforms a Kalman filter designed using the same ROM, which is attributable to the nonlinearity of its policy (see Appendix I for a quantification of this nonlinearity). Finally, the RL-ROE also yields accurate high-dimensional state estimates for reference trajectories corresponding to various parameter values without direct knowledge of the latter.

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

## A  DYNAMIC MODE DECOMPOSITION

In this appendix, we describe the DMD algorithm (Schmid, 2010; Tu et al., 2014), which is a popular data-driven method to extract spatial modes and low-dimensional dynamics from a dataset of high-dimensional snapshots. Here, we use the DMD to construct a ROM of the form (2) given an observation model $C$ and a concatenated collection of snapshots $Z_{\text{train}} = \{Z^\mu\}_{\mu \in S}$, where each $Z^\mu = \{z_0^\mu, \ldots, z_m^\mu\}$ contains snapshots from a trajectory of (1a) for a specific value $\mu$.

Fundamentally, the DMD seeks a best-fit linear model of the dynamics in the form of a matrix $A \in \mathbb{R}^{n \times n}$ such that $z_{k+1} \simeq A z_k$. First, arrange the snapshots into two time-shifted matrices

$$X = \{z_0^{\mu_1}, \ldots, z_{m-1}^{\mu_1}, \ldots, z_0^{\mu_q}, \ldots, z_{m-1}^{\mu_q}\}, \tag{15a}$$

$$Y = \{z_1^{\mu_1}, \ldots, z_m^{\mu_1}, \ldots, z_1^{\mu_q}, \ldots, z_m^{\mu_q}\}, \tag{15b}$$

where $q$ denotes the number of elements in $S$. The best-fit linear model is then given by $A = YX^\dagger$, where $X^\dagger$ is the pseudoinverse of $X$. The ROM is then obtained by projecting the matrices $A$ and $C$ onto a basis $U$ consisting of the $r$ leading left singular vectors of $X$, which approximate the $r$ leading PCA modes of $Z$. Using the truncated singular value decomposition

$$X = U\Sigma V^\mathsf{T} \tag{16}$$

where $U, V \in \mathbb{R}^{n \times r}$ and $\Sigma \in \mathbb{R}^{r \times r}$, the resulting reduced-order state-transition and observation models are given by

$$A_r = U^\mathsf{T} A U = U^\mathsf{T} Y V \Sigma^{-1}, \tag{17a}$$
$$C_r = CU. \tag{17b}$$

Conveniently, the ROM matrices $A_r$ and $C_r$ can be calculated directly from the truncated SVD of $X$, which avoids forming the large $n \times n$ matrix $A$.

## B  BAYESIAN INTERPRETATION

Here, we evaluate the meaningfulness of the RL-ROE design by framing the estimator dynamics (3) and the optimization problem (5) in the context of Bayesian inference. Unless stated otherwise, we consider the estimation problem in terms of the reduced state $x_k$, governed by the ROM (2).

### B.1  BAYESIAN OPTIMAL FILTER

From a Bayesian perspective, the goal at each time $k = 1, \ldots, K$ is to calculate $p(x_k|y_{1:k})$, the posterior probability density function (pdf) measuring our belief in the true state $x_k$ given the measurement data $y_{1:k} = \{y_1, \ldots, y_k\}$. It is assumed that we know the transition model $p(x_k|x_{k-1})$ describing the dynamics of the system, the observation model $p(y_k|x_k)$ relating state and measurement data, and the initial pdf $p(x_0)$. Then, the posterior pdf $p(x_k|y_{1:k})$ can formally be obtained recursively by alternating between prediction and update steps (Särkkä, 2013).

**Prediction step.** Starting from the posterior pdf $p(x_{k-1}|y_{1:k-1})$ at $k-1$, one first uses the dynamics of the system to compute the prior pdf at $k$ via the Chapman-Kolmogorov equation

$$p(x_k|y_{1:k-1}) = \int p(x_k|x_{k-1})p(x_{k-1}|y_{1:k-1})dx_{k-1}, \tag{18}$$

where the Markovian property of the dynamics has been used.

**Update step.** The prior is then updated using the new measurement $y_k$ via Bayes' rule, yielding the posterior pdf

$$p(x_k|y_{1:k}) = \frac{p(y_k|x_k)p(x_k|y_{1:k-1})}{p(y_k|y_{1:k-1})}, \tag{19}$$

where the normalizing constant is $p(y_k|y_{1:k-1}) = \int p(y_k|x_k)p(x_k|y_{1:k-1})dx_k$.

These equations generally do not admit analytical solutions, except in the linear and Gaussian setting which results in the Kalman filter (Särkkä, 2013). Particle filters provide an approximate solution in the general setting but they are computationally expensive, their required ensemble size scaling exponentially with the state dimension (Daum & Huang, 2003; Snyder et al., 2008).

### B.2  RL-ROE FROM A BAYESIAN PERSPECTIVE

To frame the RL-ROE in the context of Bayesian inference, we first need to relate the state estimate $\hat{x}_k$ to the posterior $p(x_k|y_{1:k})$. Specifically, we define $\hat{x}_k$ as the mean of the posterior,

$$\hat{x}_k = \mathbb{E}[x_k|y_{1:k}] = \int x_k p(x_k|y_{1:k})dx_k, \tag{20}$$

which, assuming that $p(x_k|y_{1:k})$ is known, is commonly referred to as the minimum mean square error (MMSE) estimator (Kay, 1993). Using this definition, we can now interpret the estimator

dynamics (3) and the optimization problem (5) from the perspective of the Bayesian filter equations, with the transition model $p(\boldsymbol{x}_k|\boldsymbol{x}_{k-1})$ and the observation model $p(\boldsymbol{y}_k|\boldsymbol{x}_k)$ given by (2a) and (2b).

**Prediction step.** Starting from the estimate $\hat{\boldsymbol{x}}_{k-1} = \mathbb{E}[\boldsymbol{x}_{k-1}|\boldsymbol{y}_{1:k-1}]$ at $k-1$, we take the expectation of (18) to obtain the 'prior' estimate at $k$,

$$\hat{\boldsymbol{x}}_k^- = \mathbb{E}[\boldsymbol{x}_k|\boldsymbol{y}_{1:k-1}] = \boldsymbol{A}_r\hat{\boldsymbol{x}}_{k-1}, \tag{21}$$

where we have made use of the linearity of the transition model (2a), and we have assumed that the process noise $\boldsymbol{w}_k$ is zero-mean (which is empirically observed in our examples; see Appendix ).

**Update step.** Formally, the estimate $\hat{\boldsymbol{x}}_k$ is given by (20) using the posterior pdf $p(\boldsymbol{x}_k|\boldsymbol{y}_{1:k})$ from the Bayes update (19). However, (19) cannot be solved explicitly since the RL-ROE does not evolve the full prior $p(\boldsymbol{x}_k|\boldsymbol{y}_{1:k-1})$. Instead, we make use of the well-known fact that the MMSE estimator is equivalent to minimizing the mean square error, or variance, of the posterior estimate (see, e.g., chapters 10 and 11 in Kay (1993) or chapter 2 in Särkkä (2013)). This can be seen as follows:

$$\begin{aligned}
\hat{\boldsymbol{x}}_k &= \mathbb{E}[\boldsymbol{x}_k|\boldsymbol{y}_{1:k}] \\
&= \int \boldsymbol{x}_k p(\boldsymbol{x}_k|\boldsymbol{y}_{1:k}) d\boldsymbol{x}_k \\
&= \arg\min_{\hat{\boldsymbol{x}}_k} \int ||\boldsymbol{x}_k - \hat{\boldsymbol{x}}_k||^2 p(\boldsymbol{x}_k|\boldsymbol{y}_{1:k}) d\boldsymbol{x}_k \\
&= \arg\min_{\hat{\boldsymbol{x}}_k} \mathbb{E}\left[||\boldsymbol{x}_k - \hat{\boldsymbol{x}}_k||^2|\boldsymbol{y}_{1:k}\right].
\end{aligned} \tag{22}$$

We relax the above minimization problem by restricting $\hat{\boldsymbol{x}}_k$ to a specific class of functions, as is done when deriving linear MMSE estimators. In the latter case, the optimal solution is

$$\hat{\boldsymbol{x}}_k = \hat{\boldsymbol{x}}_k^- + \boldsymbol{K}_k(\boldsymbol{y}_k - \boldsymbol{C}\hat{\boldsymbol{x}}_k^-), \tag{23}$$

where the prior estimate $\hat{\boldsymbol{x}}_k^-$ is given by (21), and $\boldsymbol{K}_k$ is obtained from the closed-form solution of (22) (Kay, 1993). We generalize this approach by considering the nonlinear form

$$\hat{\boldsymbol{x}}_k = \hat{\boldsymbol{x}}_k^- + \boldsymbol{\mu}_{\boldsymbol{\theta}_k}(\boldsymbol{y}_k, \hat{\boldsymbol{x}}_k^-), \tag{24}$$

where $\boldsymbol{\mu}_{\boldsymbol{\theta}_k}$ is a nonlinear function parameterized by $\boldsymbol{\theta}_k$, whose role is to update the prior $\hat{\boldsymbol{x}}_k^-$ using $\boldsymbol{y}_k$ in such a way to minimize the posterior variance. Then, the minimization problem (22) becomes

$$\boldsymbol{\theta}_k^* = \arg\min_{\boldsymbol{\theta}_k} \mathbb{E}\left[||\boldsymbol{x}_k - \hat{\boldsymbol{x}}_k||^2\right], \tag{25}$$

where it is implicitly assumed that the expectation is conditioned on $\boldsymbol{y}_{1:k}$.

**RL-ROE.** To obtain the estimator dynamics (3) and the optimization problem (5), we further consider that the function $\boldsymbol{\mu}_{\boldsymbol{\theta}_k}$ is stochastic and independent of time; it is therefore expressed by a stationary policy $\boldsymbol{\pi}_{\boldsymbol{\theta}}$ parameterized by $\boldsymbol{\theta}$. Then, combining the posterior estimate (24) with the prior estimate (21) gives the recursion

$$\hat{\boldsymbol{x}}_k = \boldsymbol{A}_r\hat{\boldsymbol{x}}_{k-1} + \boldsymbol{a}_k, \tag{26a}$$

$$\boldsymbol{a}_k \sim \boldsymbol{\pi}_{\boldsymbol{\theta}}(\,\cdot\,|\boldsymbol{y}_k, \hat{\boldsymbol{x}}_{k-1}), \tag{26b}$$

which is the estimator dynamics (3). The parameters $\boldsymbol{\theta}$ of the stationary policy are found by extending the minimization problem (25) to all time steps, yielding

$$\boldsymbol{\theta}^* = \arg\min_{\boldsymbol{\theta}} \mathbb{E}\left[\sum_{k=1}^K ||\boldsymbol{x}_k - \hat{\boldsymbol{x}}_k||^2\right]. \tag{27}$$

In the RL setting, this optimization problem is solved in a data-driven manner by sampling system trajectories. Thus, the expectation is taken over initial estimates $\hat{\boldsymbol{x}}_0$, initial true states $\boldsymbol{x}_0$, parameters $\mu$, estimate trajectories $\{\hat{\boldsymbol{x}}_1, \hat{\boldsymbol{x}}_2, \dots\}$ induced by $\boldsymbol{\pi}_{\boldsymbol{\theta}}$ through (3), and true trajectories of states $\{\boldsymbol{x}_1, \boldsymbol{x}_2, \dots\}$ and measurements $\{\boldsymbol{y}_1, \boldsymbol{y}_2, \dots\}$. Finally, we express the posterior variance through the reconstructed high-dimensional estimate $\hat{\boldsymbol{z}}_k = \boldsymbol{U}\hat{\boldsymbol{x}}_k$ and the high-dimensional true state $\boldsymbol{z}_k$, and we add a regularization term that penalizes the magnitude of the action $\boldsymbol{a}_k$. This gives

$$\boldsymbol{\theta}^* = \arg\min_{\boldsymbol{\theta}} \mathbb{E}\left[\sum_{k=1}^K ||\boldsymbol{z}_k - \boldsymbol{U}\hat{\boldsymbol{x}}_k||^2 + \lambda||\boldsymbol{a}_k||^2\right], \tag{28}$$

which is the optimization problem (5).

We conclude this analysis with a couple remarks. First, it demonstrates that the cost being minimized in (28) derives from the variance minimization principle underlying the definition of the MMSE estimator. Second, an important difference between the RL-ROE and the Bayesian optimal filter is that the RL-ROE does not require knowledge of the distribution of the process noise $\boldsymbol{w}_k$ and observation noise $\boldsymbol{v}_k$. Rather, it leverages knowledge of the true trajectories and corresponding measurements in the offline training phase to find the policy $\boldsymbol{\mu_\theta}$ that yields the minimum mean square error. Third, the RL solves the optimization problem (28) in a batch approach during the offline training phase, sampling entire trajectories of the estimate between each update of the policy parameters. The trained RL-ROE is then applied online in a recursive manner.

## C   KALMAN FILTER

The time-dependent Kalman filter that we use as a benchmark in this paper, KF-ROE, is based on the same ROM (2) as the RL-ROE, with identical matrices $\boldsymbol{A}_r$, $\boldsymbol{C}_r$ and $\boldsymbol{U}$. Similarly to the RL-ROE, the reduced-order estimate $\hat{\boldsymbol{x}}_k$ is given by equation (3a), from which the high-dimensional estimate is reconstructed as $\hat{\boldsymbol{z}}_k = \boldsymbol{U}\hat{\boldsymbol{x}}_k$. However, the KF-ROE differs from the RL-ROE in its definition of the action $\boldsymbol{a}_k$ in (3a), which is instead given by the linear feedback term (4). The calculation of the optimal Kalman gain $\boldsymbol{K}_k$ in (4) requires the following operations at each time step:

$$\boldsymbol{P}_k^- = \boldsymbol{A}_r\boldsymbol{P}_{k-1}\boldsymbol{A}_r^\mathsf{T} + \boldsymbol{Q}_k, \tag{29}$$

$$\boldsymbol{S}_k = \boldsymbol{C}_r\boldsymbol{P}_k^-\boldsymbol{C}_r^\mathsf{T} + \boldsymbol{R}_k, \tag{30}$$

$$\boldsymbol{K}_k = \boldsymbol{P}_k^-\boldsymbol{C}_r^\mathsf{T}\boldsymbol{S}_k^{-1}, \tag{31}$$

$$\boldsymbol{P}_k = (\boldsymbol{I} - \boldsymbol{K}_k\boldsymbol{C}_r)\boldsymbol{P}_k^-, \tag{32}$$

where $\boldsymbol{P}_k^-$ and $\boldsymbol{P}_k$ are respectively the a priori and a posteriori estimate covariance matrices, $\boldsymbol{S}_k$ is the innovation covariance, and $\boldsymbol{Q}_k$ and $\boldsymbol{R}_k$ are respectively the covariance matrices of the process noise $\boldsymbol{w}_k$ and observation noise $\boldsymbol{v}_k$ in the ROM (2). Following a standard procedure, we tune these noise covariance matrices to yield the best possible results (Simon, 2006). We assume that $\boldsymbol{Q}_k = \beta_Q\boldsymbol{I}$ and $\boldsymbol{R}_k = \beta_R\boldsymbol{I}$ and perform a line search to find the values of $\beta_Q$ and $\beta_R$ that yield the best performance. This resulted in $\beta_Q = 10^3$ and $\beta_R = 1$ for the Burgers example, and $\beta_Q = 10^9$ and $\beta_R = 1$ for the Navier-Stokes example. At time step $k = 0$, the a posteriori estimate covariance is initialized as $\boldsymbol{P}_0 = \mathrm{cov}(\boldsymbol{U}^\mathsf{T}\boldsymbol{z}_0 - \hat{\boldsymbol{x}}_0)$, which can be calculated from the distributions (10).

## D   RL ALGORITHM, HYPERPARAMETERS AND LEARNING CURVES

We employ the Proximal Policy Optimization (PPO) algorithm (Schulman et al., 2017) to find the optimal policy parameters $\boldsymbol{\theta}^*$. PPO alternates between sampling data by computing a set of trajectories $\{\tau_1, \tau_2, \tau_3, \dots\}$ using the most recent version of the policy, and updating the policy parameters $\boldsymbol{\theta}$ in a way that increases the probability of actions that led to higher rewards during the sampling phase. The policy $\boldsymbol{\pi_\theta}$ encodes a diagonal Gaussian distribution described by a neural network that maps from observation to mean action, $\boldsymbol{\mu_{\theta'}}(\boldsymbol{o}_k)$, together with a vector of standard deviations $\boldsymbol{\sigma}$, so that $\boldsymbol{\theta} = \{\boldsymbol{\theta'}, \boldsymbol{\sigma}\}$. We utilize the Stable Baselines3 (SB3) implementation of PPO (Raffin et al., 2019) and define our MDP as a custom environment in OpenAI Gym (Brockman et al., 2016).

For both the Burgers and Navier-Stokes examples, the stochastic policy $\boldsymbol{\pi_\theta}$ is trained with PPO using the default hyperparameters from Stable Baselines3, except for the discount factor $\gamma$ which we choose as $0.75$. The mean output of the stochastic policy and the value function are approximated by two neural networks, each containing two hidden layers with 64 neurons and tanh activation functions. The input to the policy is normalized using a running average and standard deviation during the training process, which alternates between sampling data for 10 trajectories (of length 200 timesteps each) and updating the policy. Each policy update consists of multiple gradients steps through the most recent data using 10 epochs, a minibatch size of $64$ and a learning rate of $0.0003$. The policy is trained for a total of one to three million timesteps, corresponding to 5000 to 15000 trajectories, which depending on the dimensionality of the ROM takes between 15 min and an hour on a Core i7-12700K CPU. Figure 9 reports the learning curves corresponding to the unforced and forced cases. During training, the policy is tested (with stochasticity switched off) after each update

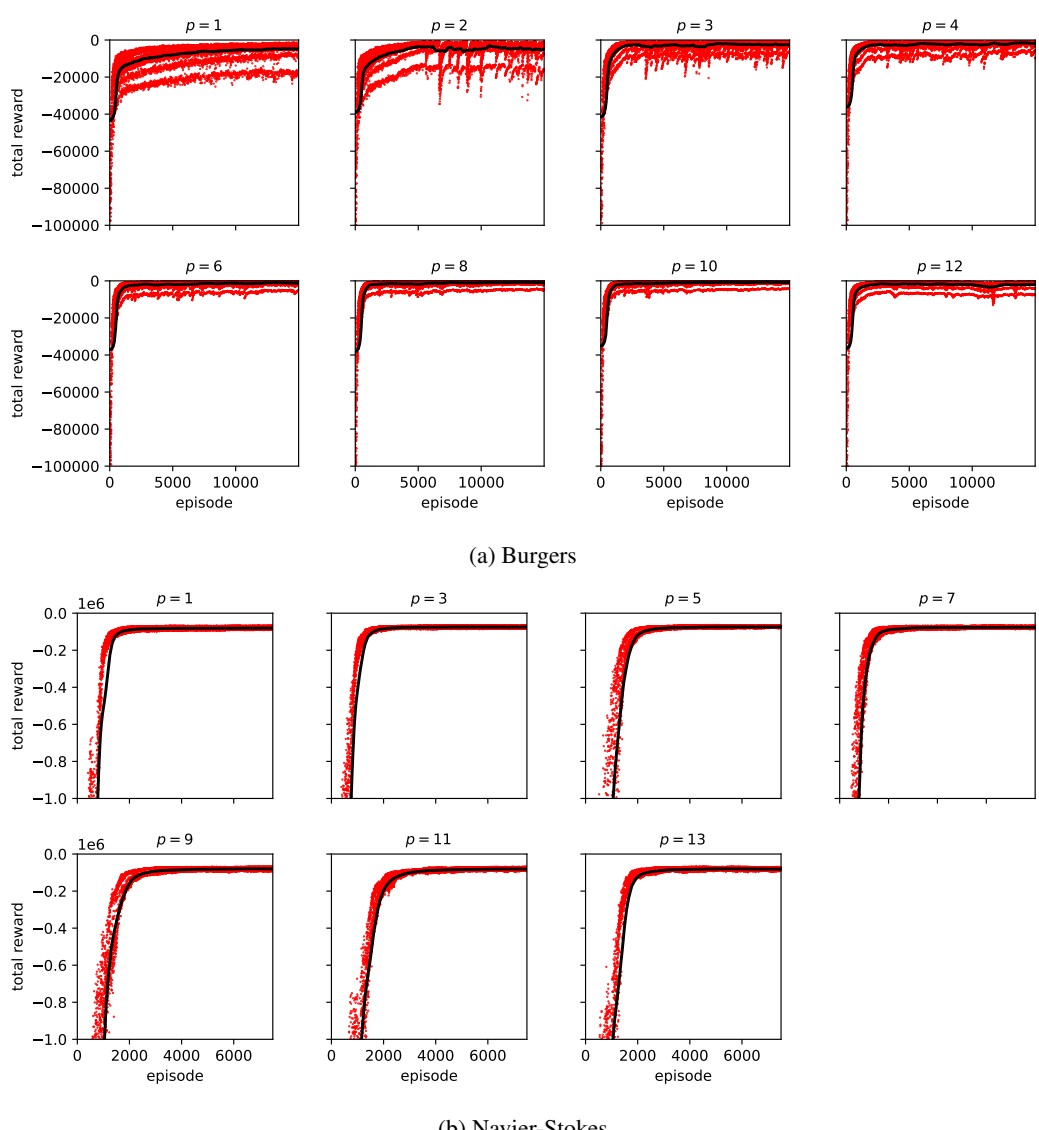

(a) Burgers

(b) Navier-Stokes

Figure 9: Learning curves for the stochastic policy of the RL-ROE for the (a) Burgers and (b) Navier-Stokes cases. Each plot corresponds to a different number $p$ of sensor measurements. The line and shaded area show the mean and standard deviation of the results over 10 runs, each one smoothed with a moving average of size 100 episodes.

using 20 separate test trajectories, and is saved if it outperforms the previous best policy. Finally, the RL-ROE is assigned the latest saved policy upon ending of the training process, and the stochasticity of the policy is switched off during subsequent evaluation of the RL-ROE.

# E    CONSTRUCTION OF THE OBSERVATION MATRIX

We describe how we construct the observation matrix $C$ in the Burgers and Navier-Stokes examples, once the number, type and locations of the sensors have been chosen.

In the Burgers example, the state vector $z_k \in \mathbb{R}^n$ contains the values of $u$ at $n$ collocation points, and the measurements $y_k \in \mathbb{R}^p$ consist of the values of $u$ at $p$ equally-spaced sensors. Let us introduce the indices $\{j_1, \ldots, j_p\}$ of the entries in $z_k$ corresponding to the measurements $y_k$. Then, $y_k$ and

$z_k$ can be related by $y_k = Cz_k$, where the matrix $C \in \mathbb{R}^{p \times n}$ contains ones at the entries indexed $\{(1, j_1), \ldots, (p, j_p)\}$, and zeros everywhere else.

In the Navier-Stokes example, the state vector $z_k \in \mathbb{R}^{2n}$ contains the horizontal and vertical components of velocity $u$ at $n$ collocation points, and the measurements $y_k \in \mathbb{R}^{2p}$ consist of the components of $u$ at $p$ equally-spaced sensors. Let us introduce the indices $\{j_1, \ldots, j_{2p}\}$ of the entries in $z_k$ corresponding to the measurements $y_k$. Then, $y_k$ and $z_k$ can be related by $y_k = Cz_k$, where the matrix $C \in \mathbb{R}^{2p \times n}$ contains ones at the entries indexed $\{(1, j_1), \ldots, (2p, j_{2p})\}$, and zeros everywhere else.

## F    ADDITIONAL RESULTS FOR THE BURGERS EQUATION

Figure 10 shows spatio-temporal contours of the reference solution and corresponding estimates, for $p = 2$ and 12. The RL-ROE vastly outperforms the KF-ROE for $p = 2$, while for $p = 12$ the KF-ROE is slightly more accurate.

## G    MODES FOR FLOW PAST A CYLINDER

The first 18 modes (i.e. columns of $U$) of the 20-dimensional ROM for the flow past a cylinder are displayed in Figure 11.

## H    EFFECT OF OBSERVATION NOISE FOR THE NAVIER-STOKES EQUATIONS

In this appendix, we evaluate the estimation accuracy of the RL-ROE in the presence of non-zero observation noise $n_k$ polluting the sensor measurements $y_k$ in (1). Specifically, we consider that $n_k$ is Gaussian white noise of standard deviation $\sigma = 0.1$. For the case of $p = 3$ sensors, Figure 12 shows the time series of the noise-free measurements contained in $y_k$ for various values of $Re$, together with their polluted counterpart (recall each sensor measures two components of velocity, as detailed in Appendix E). Using the noisy measurements, we then repeat the same experiments carried out in the main text; the results are shown in Figures 13 and 14, which are the counterparts of Figures 5 and 6. We observe excellent robustness of the RL-ROE in the presence of noise, with the estimate maintaining high accuracy.

## I    NONLINEARITY OF THE TRAINED POLICY

In this appendix, we evaluate the degree of nonlinearity of the RL-trained policies $\pi_\theta$ obtained in our Burgers and Navier-Stokes examples. Since the stochasticity of $\pi_\theta$ is switched off during online deployment, $\pi_\theta$ can be expressed by its mean function $\mu_{\theta'}$ (see Appendix D) so that the action in (3b) becomes

$$a_k = \mu_{\theta'}(y_k, \hat{x}_{k-1}). \tag{33}$$

We therefore quantify the nonlinearity of the policy by evaluating the Jacobian of the function $\mu_{\theta'}$ with respect to its two arguments $y_k$ and $\hat{x}_{k-1}$. The Jacobian is a matrix whose components are the first-order derivatives $\partial\mu_i/\partial y_j$ and $\partial\mu_i/\partial\hat{x}_j$, where $(i, j)$ refers to the indices of the vectors entries in $\mu$ and $y_k$ or $\hat{x}_{k-1}$, respectively. Instead of looking at individual components, we consider the Frobenius norm of the Jacobian, defined as

$$\left\| \frac{\partial\mu_{\theta'}}{\partial(y, \hat{x})} \right\|_F^2 = \sum_{i=1}^r \left[ \sum_{j=1}^p \left( \frac{\partial\mu_i}{\partial y_j} \right)^2 + \sum_{j=1}^r \left( \frac{\partial\mu_i}{\partial\hat{x}_j} \right)^2 \right]. \tag{34}$$

The Jacobian (and its norm) of a linear policy will be independent of the input values, while the Jacobian (and its norm) of a nonlinear policy will change with the input values. Figure 15 show the distribution of the norm of the Jacobian of the trained policies obtained in the Burgers and Navier-Stokes examples for various values of $p$. The distributions are obtained by calculating the Jacobian along a solution trajectory of the RL-ROE corresponding to the results shown in Sections 4.1 and 4.2. The wide spread of the distributions demonstrates that the mean function $\mu_{\theta'}$ trained by the RL process is highly nonlinear, as opposed to the linear correction term (4) in the KF-ROE.

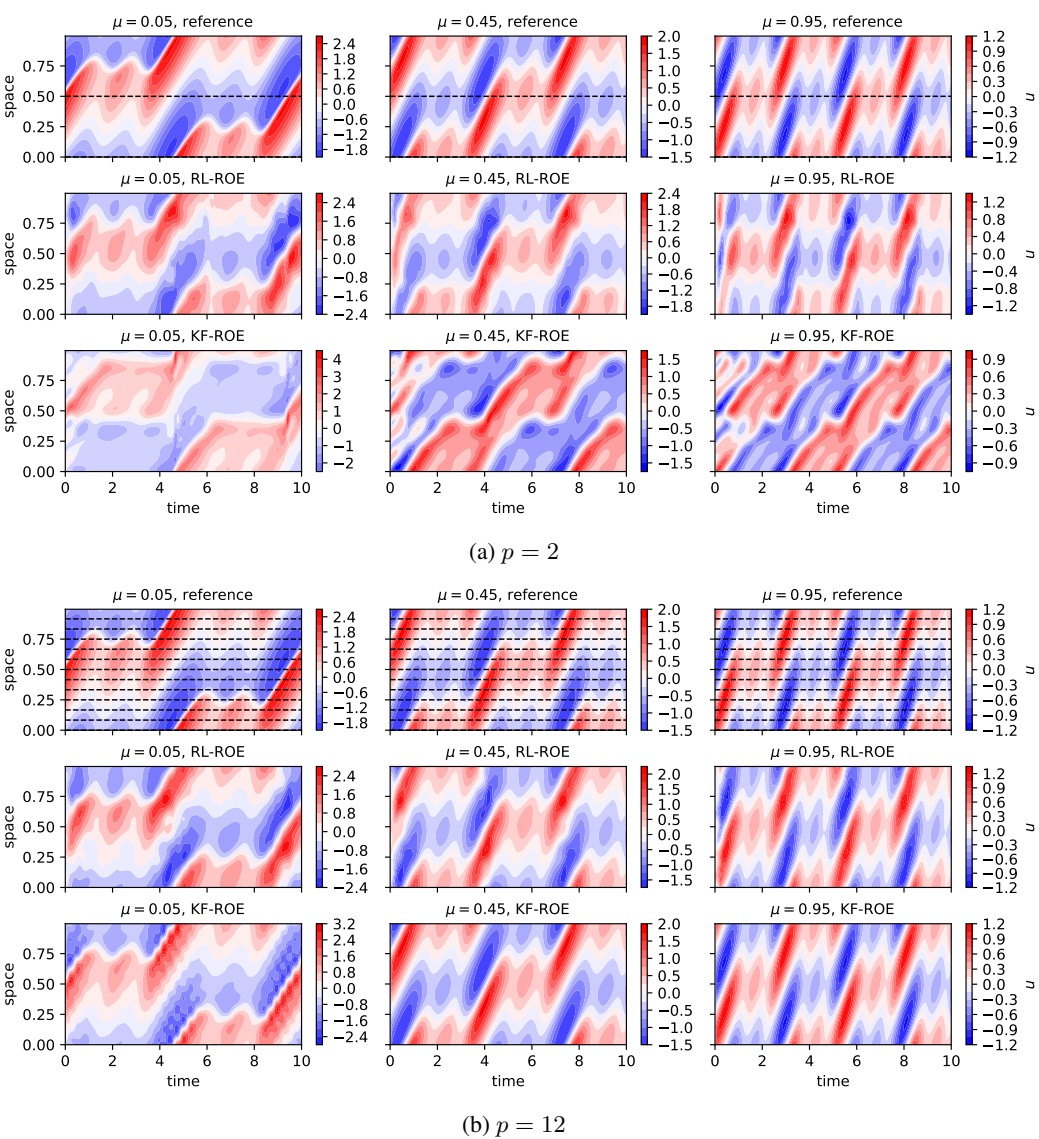

Figure 10: Burgers equation with (a) $p = 2$ and (b) $p = 12$ sensors. Reference trajectories for values of $\mu$ not seen during training and corresponding RL-ROE and KF-ROE estimates. The dashed lines on the reference trajectory plots indicate the sensor data seen by the RL-ROE and KF-ROE.

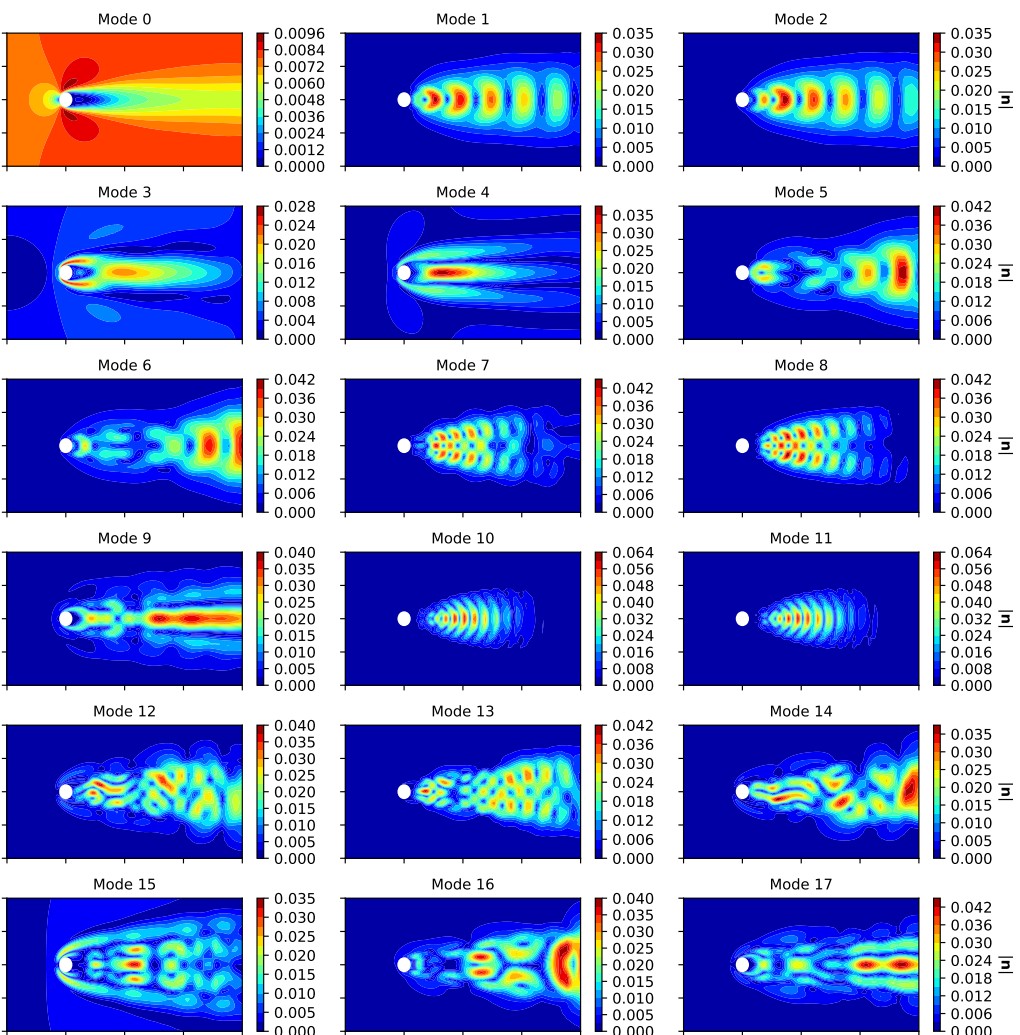

Figure 11: First 18 modes of the 20-dimensional ROM for the flow past a cylinder.

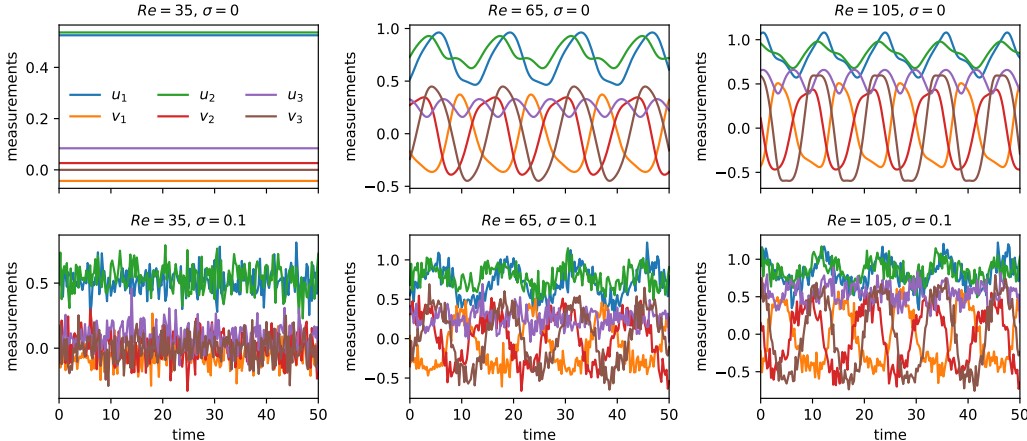

Figure 12: Navier-Stokes equations with $p = 3$ sensors and observation noise of std $\sigma$. Time series of measurements for different values of $Re$ for $\sigma = 0$ (top row) and $\sigma = 0.1$ (bottom row).

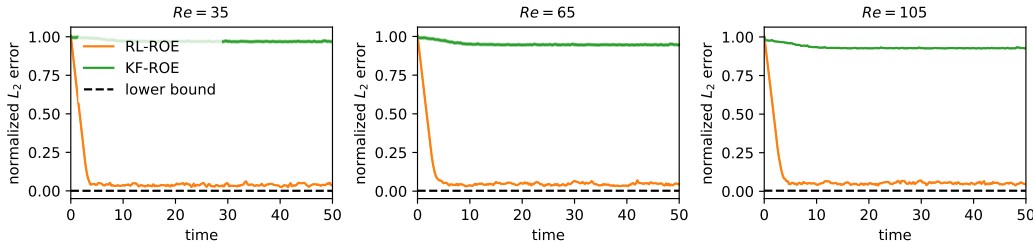

Figure 13: Navier-Stokes equations with $p = 3$ sensors and observation noise of std $\sigma = 0.1$. Normalized $L_2$ error of the RL-ROE and KF-ROE for the estimation of trajectories corresponding to values of $Re$ not seen during training.

## J  DISTRIBUTION OF PROCESS NOISE

We evaluate empirically the distribution of the process noise $\boldsymbol{w}_k$ in the ROM dynamics (2a). First, note that the vector $\boldsymbol{w}_k$ can be evaluated as $\boldsymbol{w}_k = \boldsymbol{A}_r \boldsymbol{x}_{k-1} - \boldsymbol{x}_k$, where $\boldsymbol{x}_{k-1} = \boldsymbol{U}^\top \boldsymbol{z}_{k-1}$ and $\boldsymbol{x}_k = \boldsymbol{U}^\top \boldsymbol{z}_k$ are the reduced-order projections of two consecutive states $\boldsymbol{z}_{k-1}$ and $\boldsymbol{z}_k$ solving the high-dimensional dynamics (1). For a given ROM, we can then sample the process noise by evaluating values of $\boldsymbol{w}_k$ along trajectories of the high-dimensional dynamics (1). We show in Figure 16 the distributions of the components of $\boldsymbol{w}_k$ for the Burgers and Navier-Stokes examples, using the ROM constructed in Sections 4.1 and 4.2. The sampling is done along trajectories of (1) corresponding to different parameter values (the parameter being $\mu$ for Burgers, $Re$ for Navier-Stokes), and the corresponding distributions of process noise are shown separately for each parameter value. (Note that the same ROM is shared across all parameter values.) The distributions reveal that the process noise is non-Gaussian but approximately zero-mean.

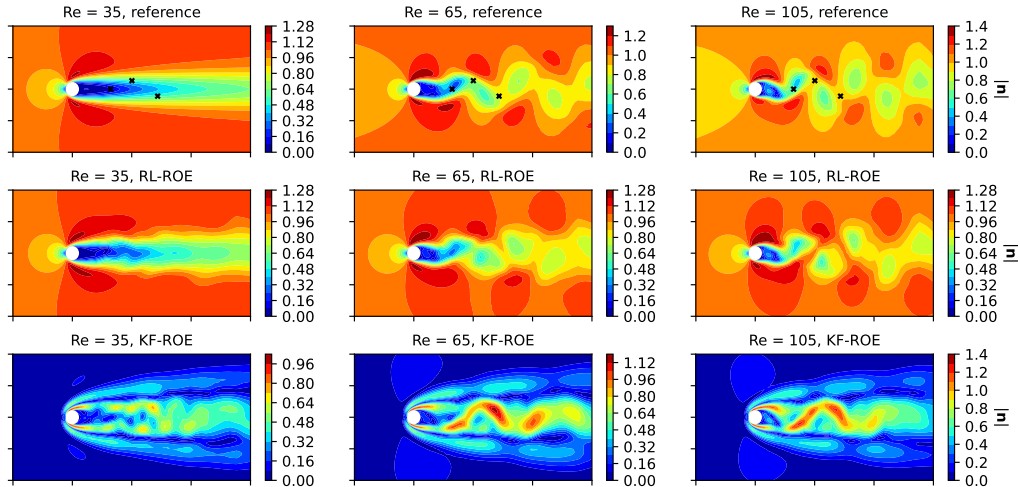

Figure 14: Navier-Stokes equations with $p = 3$ sensors and observation noise of std $\sigma = 0.1$. Velocity magnitude at $t = 50$ of the reference trajectories for values of $Re$ not seen during training and corresponding RL-ROE and KF-ROE estimates. The black crosses in the contours of the reference solutions indicate the sensor locations.

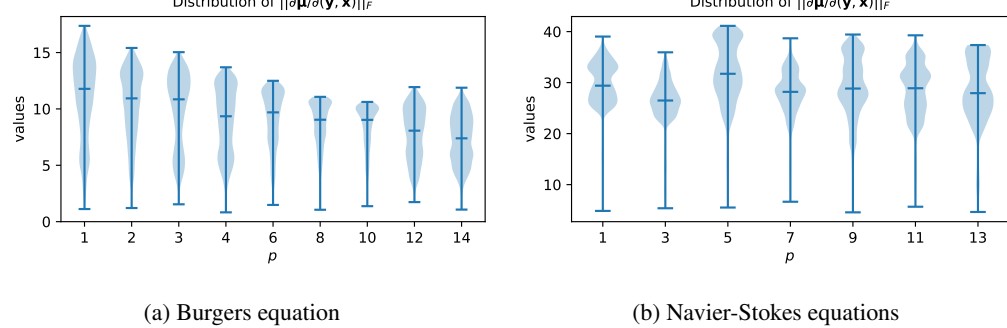

(a) Burgers equation              (b) Navier-Stokes equations

Figure 15: Nonlinearity of the trained policies obtained in the (a) Burgers and (b) Navier-Stokes examples for various $p$. Distribution of the Frobenius norm of the Jacobian of the trained mean policy $\boldsymbol{\mu}_{\theta'}$, sampled along solution trajectories of the RL-ROE.

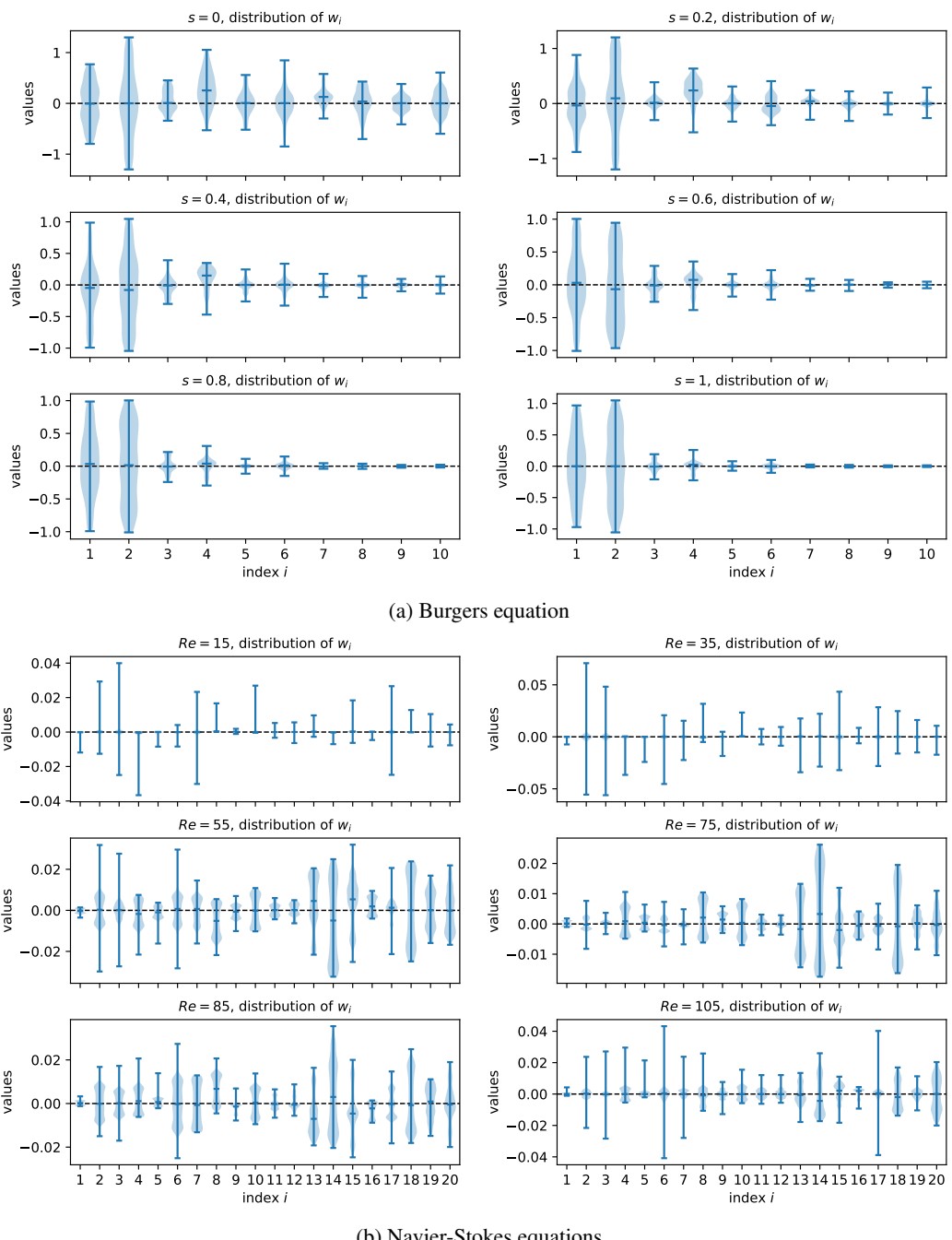

(a) Burgers equation

(b) Navier-Stokes equations

Figure 16: Distribution of process noise in the (a) Burgers and (b) Navier-Stokes examples, sampled along solution trajectories of the high-dimensional dynamics (1). The distributions of each component of $\boldsymbol{w}_k$ obtained for each parameter value are shown individually.

