# OpenReview forum: "Reinforcement Learning-Based Estimation for Partial Differential Equations"
_ICLR.cc/2023/Conference — Submitted to ICLR 2023_

### Official Review · Reviewer_rh8q · 2022-10-23

**Confidence:** 4
**Correctness:** 4
**Technical Novelty And Significance:** 3
**Empirical Novelty And Significance:** 3
**Recommendation:** 6

**Clarity, Quality, Novelty And Reproducibility:**

clear, high quality, pending my evaluation of originality on the elaboration of the last paragraph in "Related Work".

**Strength And Weaknesses:**

Strengths: interesting idea, justified approach, good simulations, well written paper.

Weaknesses/Limitations are reasonably addressed by the authors.

a few comments:
1. Eq7. lhs depend on t+1, while rhs doesn't not depend on $t+1$

2. in NAVIER-STOKES EQUATION: the dimensionality of state (the discretized solution) is huge. could you elaborate on the sensors in this case? does the sensor matrix, C, include only 1 non zero entries?  I suggest to explicitly state the dimensions of the sensor in the experiments.

3. could you elaborate on "However, this approach would not work for our purpose since the RL-ROE cannot be restricted to estimating a single or a small finite number of reference trajectories. Instead, by augmenting the state with entire snapshots from the reference trajectory instead of just a phase variable, we ensure that there is no theoretical limit to the number of reference trajectories that the policy πθ can learn"? I don't understand why including the snapshot to the state allows the policy to learn any number of trajectories. The state is augmented with $\mu$, which I thought, is enough for the generalization to unseen trajectories.

**Summary Of The Paper:**

this paper proposes to estimate ROM from samples by formulating the correction term to be a non-linear policy, which is trained by RL. The problem formulation requires a stationary MDP model, which is constructed by augmenting the state with the original $z_k$ and $\mu$.

**Summary Of The Review:**

well-written paper. the background and limitations are well explained. numerical simulations confirm the claims.

---

> ### Author Response · Authors · 2022-11-17
> **Response to Reviewer rh8q**
>
> We thank the reviewer for finding our paper interesting, well-written and of high quality. Please find below our answers to the reviewer's questions. We would also be grateful if the reviewer could mention whether they desire other aspects of the paper to be improved in order to warrant a higher score.
>
> > 1. Eq7. lhs depend on $t+1$, while rhs doesn't not depend on $t+1$
>
> The RHS of the reward function depends on $\mathbf{s}\_{k+1}$ through $\hat{\mathbf{x}}\_k$ since $s\_{k+1} = (\hat{\mathbf{x}}\_k, \mathbf{z}\_{k+1}, \mu)$
>
> > 2. in NAVIER-STOKES EQUATION: the dimensionality of state (the discretized solution) is huge. could you elaborate on the sensors in this case? does the sensor matrix, C, include only 1 non zero entries? I suggest to explicitly state the dimensions of the sensor in the experiments.
>
> We have followed the reviewer's suggestion in our revised submission, where we now explicitly state the dimension of the sensor measurements $\mathbf{y}\_k$ and we explain the construction of the observation matrix $\mathbf{C}$ in a new appendix E.
>
> In the Navier-Stokes example, each sensor measures the horizontal and vertical components of velocity at a single point. Thus, with $p$ sensors, the vector of measurements $\mathbf{y}\_k$ will contain $2p$ entries. To explain the construction of the sensor matrix $\mathbf{C}$, let us introduce the indices $\{j_1,\dots,j_{2p}\}$ of the entries in $\mathbf{z}\_k$ corresponding to the measurements $\mathbf{y}\_k$. Then, $\mathbf{y}\_k$ and $\mathbf{z}\_k$ can be related by $\mathbf{y}\_k = \mathbf{C} \mathbf{z}\_k$, where the matrix $\mathbf{C} \in \mathbb{R}^{2p \times n}$ contains ones at the entries indexed $\{(1,j_1),\dots,(2p,j_{2p})\}$, and zeros everywhere else. Thus, the matrix $\mathbf{C}$ is sparse and contains $2p$ non-zero entries.
>
> > 3. could you elaborate on "However, this approach would not work for our purpose since the RL-ROE cannot be restricted to estimating a single or a small finite number of reference trajectories. Instead, by augmenting the state with entire snapshots from the reference trajectory instead of just a phase variable, we ensure that there is no theoretical limit to the number of reference trajectories that the policy $\boldsymbol{\pi}_{\boldsymbol{\theta}}$ can learn"? I don't understand why including the snapshot to the state allows the policy to learn any number of trajectories. The state is augmented with $\mu$, which I thought, is enough for the generalization to unseen trajectories.
>
> We thank the reviewer for this comment, which led us to clarify this paragraph in the revised submission. In the context of our current work where the policy learns a single post-transient trajectory for every value of $\mu$, we agree with the reviewer that augmenting the MDP state with $\mu$ as well as time is sufficient. However, we would like our method to work for dynamical systems that possess multiple post-transient trajectories at each value of $\mu$ (for instance, systems with a double-well potential have two separate attractors in phase space). In this case, the knowledge of $\mu$ and $k$ is not enough to define the state transition model in the MDP, which is why we include the entire snapshot $\mathbf{z}_k$ of the state.

---

> ### Author Response · Authors · 2022-12-13
> **Follow up with Reviewer rh8q**
>
> We once again thank the reviewer for their enthusiastic comments and relevant questions that lead to several clarifications. We would be grateful if the reviewer could let us know if their initial concerns have been addressed, and we would appreciate any further questions or comments.

---

### Official Review · Reviewer_AsBn · 2022-10-24

**Confidence:** 4
**Correctness:** 3
**Technical Novelty And Significance:** 2
**Empirical Novelty And Significance:** 3
**Recommendation:** 5

**Clarity, Quality, Novelty And Reproducibility:**

- In my opinion, the submission is well-written and easy to follow.
- The originality of the work is somehow limited as the principle of learning estimators using reinforcement learning is well known.
- The experiments are fair and interesting instantiations of the problem.
- The presented algorithm is easy to implement and, therefore, probably easily reproducible.


**Details Of Ethics Concerns:**

-

**Strength And Weaknesses:**

The paper discusses in my opinion a very timely topic, state estimation for PDEs. The setup as such is interesting and has probably many application scenarios. The use of a dimensionality reduction method is a clever way to make the state estimation problem tractable. My biggest concern is, however,  that the algorithmic development is very heuristic. The paper misses the principled and "modern viewpoint" of Bayesian inference to model the state estimation problem, see, e.g., [1] for an introduction, and [2,3] for mathematical treatments.
Hence, the proposed mean estimator in equation (3) is arbitrarily chosen, which resembles the usual structure of the Kushner-Startonovich equation, including the prior dynamics and the innovation process.
For me, it makes sense that the reinforcement learning-based estimator outperforms the linear Kalman filter, as the estimator is based on a Gaussian error assumption. However, what is missing in such a case is a comparison, to, e.g., a nonlinear particle filter, which probably will easily outperform the proposed RL algorithm.
Another problematic aspect is that the controller only has access to the new observation and the posterior mean estimate. Even when assuming the linear prior dynamics, the innovation process is still dependent on the whole posterior distribution at the previous time step. This is justified in the remark saying that the history $h_k$ represented by $o_k$ is an incomplete summary and therefore sufficient. I would highly object that this is enough for state estimation since even in the linear quadratic case the optimal estimator is dependent on the variance of the posterior distribution (solution to the Ricatti equation).
Otherwise, I think the experiments are fair and convincing for the considered setup.

As an aside, which is probably out of scope for this work, I would argue that a problematic aspect of the setup is that a time discretization is carried out before the inference. This results in an easier-to-model discrete-time system, however, a continuous-time description is usually much more sensible in these scenarios, see, e.g., [4] for an accessible introduction and [2] for a mathematical treatment. As oftentimes the time-discretization as chosen by the Runge-Kutta integrator is selected arbitrarily to model the prior dynamics, however, the posterior dynamics often require a different time-discretization. Maybe for future work, the authors can consider this setup.

Summary:

Strong points:
- The submission is timely and the setup is very interesting.
- The submission is fairly well-written and technically correct.


Weak points:
- algorithmic development is heuristic
- Bayesian description is missing
- estimator is chosen arbitrarily and not optimal
- a simple comparison to a particle filter algorithm is missing


[1] Särkkä, Simo. Bayesian filtering and smoothing. No. 3. Cambridge university press, 2013.
[2] Elliott, Robert J., Lakhdar Aggoun, and John B. Moore. Hidden Markov models: estimation and control. Vol. 29. Springer Science & Business Media, 2008.
[3] Bain, Alan, and Dan Crisan. Fundamentals of stochastic filtering. Vol. 3. New York: Springer, 2009.
[4] Särkkä, Simo, and Arno Solin. Applied stochastic differential equations. Vol. 10. Cambridge University Press, 2019.

**Summary Of The Paper:**

The paper considers state space estimation by optimal filtering for latent partial differential equations (PDE). For this, a reinforcement-learning-based estimator is proposed.

The paper considers a given space-time discretization method of the PDE in question, resulting in a high-dimensional system of time-discretized nonlinear ordinary differential equations (ODEs). The resulting high-dimensional system is reduced in dimensionality using a linear dimensionality reduction method, dynamic mode decomposition (DMD). The state estimation problem is therefore transferred to the low-dimensional linear space.
For solving the resulting optimal filtering problem an estimator for the posterior mean of the Bayesian filtering problem is proposed. The authors consider for this an estimator which includes the low-dimensional linear dynamics and an additive nonlinear correction term (innovation process) based on the measurements.
The optimal filter is learned using a reinforcement learning (RL) algorithm, proximal policy optimization (PPO). The target of the RL algorithm is a reward parametrized by the negative squared estimation error and a penalty for the innovation gain.
The final algorithm is evaluated for two synthetic PDE filtering problems.

**Summary Of The Review:**

The paper is well-written and the setup is interesting. The principle behind the estimation algorithm presented is known. The biggest downside of this paper is, that the algorithm is developed heuristically and consequently a comparison to other nonlinear state space estimators is missing. The experiments considered are interesting and the evaluation of the considered setup is fair.

Altogether I come to the conclusion that this paper is between the ratings of 3 (reject, not good enough) and 5 (marginally below the acceptance threshold).

Some questions I had for the authors:
- What is the motivation behind the considered estimator in equation (3), why is not an estimator considered, which is fully parameterized by a neural network, i.e., $\hat{x}_k=a_k$
- Is this modeling of one linear dynamical system efficient, did you consider using a switching dynamical system as a model, e.g., [5]?
- Could this setup be extended to nonlinear dimensionality reduction methods, if anyway later a nonlinear estimator is used?
- The estimator in equation (3) assumes a zero mean noise process $w_k$. How is the empirical distribution for $w_k$ looking, is it zero mean?
- In equation (7), the hyper-parameter $\lambda_2$ should be dependent on the observation noise, i.e., if the observation noise is high $\lambda_2$ should be chosen large, as this would diminish the gain on the innovation term ($a_k$) in equation (3). How was the observation noise chosen in the experiments? Can you see an influence of the two hyper-parameters, noise variance, and $\lambda_2$?

[5] Linderman, Scott, et al. "Bayesian learning and inference in recurrent switching linear dynamical systems." Artificial Intelligence and Statistics. PMLR, 2017.

---

> ### Author Response · Authors · 2022-11-19
> **Response to Reviewer AsBn (part 1)**
>
> We thank the reviewer for finding our paper interesting and well-written. Please find below our answers to the reviewer's questions, which lead to substantial additions in the revised version and helped strengthen the paper. We hope that the reviewer will consider raising their score in light of the new additions and clarifications.
>
> > algorithmic development is heuristic
>
> We agree that the algorithmic development was heuristic in our original submission. Following the reviewer's comment, we present in a new Appendix B of the revised paper a principled derivation of our estimator from the perspective of Bayesian inference, using results from minimum mean square error (MMSE) estimation. Furthermore, we now state explicitly in eq (5) the optimization problem solved by RL to find the policy parameters. We also note that designing state estimators by correcting the dynamics model with a measurement-dependent term is a standard approach in control theory, where they are called state observers [1].
>
> The Bayesian interpretation provides some insight into the meaning of our estimator dynamics and optimization problem. As the reviewer has noted, our estimate $\hat{\mathbf{x}}\_k$ at time $k$ can be related to the mean of the posterior pdf $p(\mathbf{x}\_k|\mathbf{y}\_{1:k})$ of the true state given the history of measurements $\mathbf{y}\_{1:k} = \{\mathbf{y}\_1,\dots,\mathbf{y}\_k\}$. This is also known as the minimum mean square error (MMSE) estimator, which is equivalent to minimizing the variance of the posterior estimate [2,3]. In other words, $\hat{\mathbf{x}}\_k = \mathbb{E}[\mathbf{x}\_k|\mathbf{y}\_{1:k}]$ also minimizes $\mathbb{E}[||\mathbf{x}\_k-\hat{\mathbf{x}}\_k||^2]$ conditioned on $\mathbf{y}\_{1:k}$. By expressing $\hat{\mathbf{x}}\_k = \hat{\mathbf{x}}\_k^- + \boldsymbol{\mu}\_{\boldsymbol{\theta}} (\hat{\mathbf{x}}\_k^-,\mathbf{y}\_k)$, where the prior $\hat{\mathbf{x}}\_k^-$ is obtained by propagating the posterior $\hat{\mathbf{x}}\_{k-1}$ at previous time $k-1$ through the system dynamics, the problem is now to find the parameters $\boldsymbol{\theta}$ of the correction function $\boldsymbol{\mu}\_{\boldsymbol{\theta}}$ so that the posterior variance is minimized. Owing to the stationarity of $\boldsymbol{\mu}\_{\boldsymbol{\theta}}$, we can extend this minimization problem to all time steps, resulting in the minimization problem that we solve with RL in eq (5). This derivation reveals that the cost function derives from the variance minimization principle underlying the definition of the MMSE estimator.
>
> [1] Gildas Besançon. Nonlinear observers and applications, volume 363. Springer, 2007.
>
> [2] Simo Sarkka. Bayesian filtering and smoothing. Cambridge university press, 2013.
>
> [3] Steven M Kay. Fundamentals of statistical signal processing: estimation theory. Prentice-Hall, Inc., 1993.
>
> > Bayesian description is missing
>
> We have added a Bayesian description of our estimator in the revised version; see our answer above.
>
> > estimator is chosen arbitrarily and not optimal
>
> In the revised version, we motivate the design of our estimator by deriving its dynamics, as well as the optimization problem defining the policy parameters, in the framework of Bayesian inference; see our answer above. Our estimator is therefore not chosen arbitrarily. Although we recognize that some simplifications result in a sub-optimal estimator, we do not think that the lack of global optimality of our estimator constitutes a weakness of our work, since a computationally tractable optimal estimator for nonlinear systems does not exist.

---

> ### Author Response · Authors · 2022-11-19
> **Response to Reviewer AsBn (part 2)**
>
> We thank the reviewer for finding our paper interesting and well-written. Please find below our answers to the reviewer's questions, which lead to substantial additions in the revised version and helped strengthen the paper. We hope that the reviewer will consider raising their score in light of the new additions and clarifications.
>
> > Another problematic aspect is that the controller only has access to the new observation and the posterior mean estimate. Even when assuming the linear prior dynamics, the innovation process is still dependent on the whole posterior distribution at the previous time step. This is justified in the remark saying that the history $h_k$ represented by $o_k$ is an incomplete summary and therefore sufficient. I would highly object that this is enough for state estimation since even in the linear quadratic case the optimal estimator is dependent on the variance of the posterior distribution (solution to the Ricatti equation).
>
> We agree with the reviewer that a classical estimator, which calculates its estimate online in real time, needs to compute an approximation of the posterior distribution besides just the mean. However, our approach differs from the classical setting in the sense that our estimator is first trained offline in a data-driven manner, during which we sample many trajectories of the true state and corresponding measurement. The trained estimator has therefore learned much prior information that is inaccessible to classical estimators directly deployed online without offline training.
>
> While we recognize in the paper that our design is sub-optimal given that we substitute the entire history $\mathbf{h}_k$ with only the current observation $\mathbf{o}_k$, we observe empirically that our setup leads to an estimator that performs well in practice, as acknowledged by the reviewer. Since we do not claim that our estimator is optimal, we do not think that the lack of global optimality of the design should constitute a weakness. As mentioned above, a computationally tractable optimal estimator for nonlinear systems does not exist, and even less one that works in high-dimensional settings involving parameter uncertainty and dynamical bifurcations.
>
> > a simple comparison to a particle filter algorithm is missing
>
> While particle filters seem at first an attractive option for our setting, they are in fact not suited to our setup for two reasons.
>
> The first reason is that, in our case, the process noise $\mathbf{w}_k$ accounts for both the unmodeled nonlinear dynamics (due to the linearity of the ROM) and the parameter uncertainty (due to the same ROM being used for estimating trajectories corresponding to different parameter values). However, its distribution is unknown and highly non-Gaussian (see the newly created Appendix J). It is therefore not clear how to evolve the particles to obtain the prior during the prediction step: either the process noise is assumed to be Gaussian and the particle filter might not perform better (considering its own complexities) than a regular Kalman filter, or the process noise is modeled explicitly. Since the process noise partly originates from the parametric uncertainty, this second option directly relates to incorporating robustness into particle filters, which is a current area of research.
>
> The second, and major, reason has to do with the computational cost of particle filters. Since they approximate a pdf in $n$ dimensions with a weighted set of particles, the number of particles required grows exponentially with $n$ [4,5]. Therefore, particle filters demand large computational resources and they are in practice not applicable to systems of dimension $n$ larger than about 10 in the context of embedded control systems that we have in mind. We note that the ROM in our Navier-Stokes example of flow past a cylinder has 20 dimensions, and more complex dynamics would require ROMs of even larger dimensions.
>
> [4] Fred Daum and Jim Huang. Curse of dimensionality and particle filters. In 2003 IEEE Aerospace
> Conference Proceedings (Cat. No. 03TH8652), volume 4, pp. 4 1979–4 1993. IEEE, 2003.
>
> [5] Chris Snyder, Thomas Bengtsson, Peter Bickel, and Jeff Anderson. Obstacles to high-dimensional
> particle filtering. Monthly Weather Review, 136(12):4629–4640, 2008.

---

> ### Author Response · Authors · 2022-11-19
> **Response to Reviewer AsBn (part 3)**
>
> We thank the reviewer for finding our paper interesting and well-written. Please find below our answers to the reviewer's questions, which lead to substantial additions in the revised version and helped strengthen the paper. We hope that the reviewer will consider raising their score in light of the new additions and clarifications.
>
> > What is the motivation behind the considered estimator in equation (3), why is not an estimator considered, which is fully parameterized by a neural network, i.e., $\hat{x}_k = a_k$
>
> The motivation behind including the dynamic model $\mathbf{A}$ in the estimator dynamics $\hat{\mathbf{x}}\_k = \mathbf{A} \hat{\mathbf{x}}\_{k-1} + \boldsymbol{\mu}\_{\boldsymbol{\theta}}(\mathbf{y}\_k, \hat{\mathbf{x}}\_{k-1})$ is that it will improve the accuracy and robustness of the estimator in situations where the dynamic model is accurate, since the role of $\boldsymbol{\mu}\_{\boldsymbol{\theta}}$ is simply to correct, or update, the value of the estimate using the most recent measurement data $\mathbf{y}\_k$. This motivation finds rigorous justification in the newly added recursive Bayesian interpretation of the estimator. Indeed, the best recursive linear estimator for $\mathbf{x}\_{k}$ using the measurement data up to time $k$ is given by $\hat{\mathbf{x}}\_{k} = \hat{\mathbf{x}}\_k^- + \mathbf{K}\_k(\mathbf{y}\_k - \mathbf{C} \hat{\mathbf{x}}\_k^-)$ (the update step of the Bayesian filter). Here, $\hat{\mathbf{x}}\_k^-$ is the prior estimate of $\mathbf{x}\_k$ using the measurement data up to time $k-1$, and is obtained by propagating the estimate at previous time $\hat{\mathbf{x}}\_{k-1}$ through the dynamic model; that is, $\hat{\mathbf{x}}\_k^- = \mathbf{A} \hat{\mathbf{x}}\_{k-1}$ (the prediction step of the Bayesian filter). Combining these two steps shows that the best linear estimator for $\hat{\mathbf{x}}\_{k}$ is of the form $\hat{\mathbf{x}}\_{k} = \mathbf{A} \hat{\mathbf{x}}\_{k-1} + \boldsymbol{\mu}\_{\boldsymbol{\theta}}(\mathbf{y}\_k, \hat{\mathbf{x}}\_{k-1})$, which is the same as we have considered in eq (3).
>
> > Is this modeling of one linear dynamical system efficient, did you consider using a switching dynamical system as a model, e.g., [5]?
>
> Using a switching dynamical system as a model is indeed an attractive option to handle complex nonlinear dynamics and parametric variations. How to properly integrate them within our RL-based framework is, however, not immediately clear and a research problem in itself. Therefore we have not considered them in the present work, but we do intend to explore such a direction in the future.
>
> > Could this setup be extended to nonlinear dimensionality reduction methods, if anyway later a nonlinear estimator is used?
>
> This is correct. Since we use a nonlinear estimator, our framework could in principle be directly extended to nonlinear ROMs, a direction that we are currently pursuing.
>
> > The estimator in equation (3) assumes a zero mean noise process $w_k$. How is the empirical distribution for $w_k$ looking, is it zero mean?
>
> We agree with the reviewer that our estimator assumes a zero-mean process noise $\mathbf{w}_k$ (now stated explicitly in Appendix B of the revised submission), which we should verify empirically. Following the reviewer's suggestion, we have plotted in Appendix I of the revised submission the empirical distribution of the process noise for the Burgers and Navier-Stokes examples. We observe that the distributions of the components of $\mathbf{w}_k$ are approximately zero-mean, which is consistent with our estimator.
>
> > In equation (7), the hyper-parameter $\lambda_2$ should be dependent on the observation noise, i.e., if the observation noise is high $\lambda_2$ should be chosen large, as this would diminish the gain on the innovation term ($a_k$) in equation (3). How was the observation noise chosen in the experiments? Can you see an influence of the two hyper-parameters, noise variance, and $\lambda_2$?
>
> This is an excellent point. Our initial aim in including $\lambda_2$ was precisely to improve the robustness of the estimator in the presence of observation noise. In the original submission, we did not consider measurement noise so we naturally selected $\lambda_2 = 0$. Following the reviewer's suggestion, we revisited our Navier-Stokes experiment with white Gaussian measurement noise, and we studied the interplay between noise variance, $\lambda_2$, and estimation error. We found that the estimation error increases with the measurement noise (as expected), but we also found that it is more or less stable for values of $\lambda_2$ below 1, and increases for values beyond 1. A detailed study of the precise effect of $\lambda_2$ on the estimation performance should therefore be left to future work. In a new Appendix H, we present the results obtained for the Navier-Stokes experiment in the presence of observation noise, showing strong robustness of the estimator with respect to noise even with $\lambda_2 = 0$.

---

> ### Author Response · Authors · 2022-11-19
> **Response to Reviewer AsBn (part 4)**
>
> We thank the reviewer for finding our paper interesting and well-written. Please find below our answers to the reviewer's questions, which lead to substantial additions in the revised version and helped strengthen the paper. We hope that the reviewer will consider raising their score in light of the new additions and clarifications.
>
> > As an aside, which is probably out of scope for this work, I would argue that a problematic aspect of the setup is that a time discretization is carried out before the inference. This results in an easier-to-model discrete-time system, however, a continuous-time description is usually much more sensible in these scenarios, see, e.g., [4] for an accessible introduction and [2] for a mathematical treatment. As oftentimes the time-discretization as chosen by the Runge-Kutta integrator is selected arbitrarily to model the prior dynamics, however, the posterior dynamics often require a different time-discretization. Maybe for future work, the authors can consider this setup.
>
> We have clarified in the updated submission that the discrete-time dynamics expressed in eq (1) are the result of a high-fidelity numerical discretization of the PDE, and can therefore be conceptually considered `exact'. The time interval $\Delta t$ separating two discrete time steps $k$ and $k+1$ is not the time step used in the time integrator; getting from $\mathbf{z}\_k$ to $\mathbf{z}\_{k+1}$ might involve a large number of iterations of the time integrator. For example, the DMD algorithm that we use to construct our discrete-time ROM is trained on high-fidelity trajectories of the PDE, obtained from a temporal discretization of the PDE using a much smaller time step than that corresponding to the discrete-time ROM (see Section 4).
>
> > The originality of the work is somehow limited as the principle of learning estimators using reinforcement learning is well known.
>
> We would be very interested if the reviewer could refer us to some well known papers learning estimators using reinforcement learning, as our literature search did not return many results.

---

> ### Author Response · Authors · 2022-12-13
> **Follow up with Reviewer AsBn**
>
> We once again thank the reviewer for their many suggestions that lead to several additions and improvements. We would be grateful if the reviewer could let us know if their initial concerns have been addressed, and we would appreciate any further questions or comments.

---

### Official Review · Reviewer_TCZP · 2022-10-24

**Confidence:** 3
**Correctness:** 3
**Technical Novelty And Significance:** 3
**Empirical Novelty And Significance:** 3
**Recommendation:** 6

**Clarity, Quality, Novelty And Reproducibility:**

As mentioned above, the paper is clearly written. I have two main comments:

(1) To what extent is the proposed method specific to PDEs? It seems that once the PDE is discretized (which is done separately from the proposed method), it is a quite generic method that is developed and applied. In that case, it might be interesting to adjust accordingly the presentation of the paper.

(2) Are there other (better) baselines than a simple Kalman filter? Or could you please provide more details on why this is a good baseline?


**Strength And Weaknesses:**

Strength: the paper is clearly written and the ideas are well presented.

Weaknesses: (1) it seems that the framework and the model purely deal with a discrete time and finite dimensional system. The connection with PDEs (mentioned in the title) is not clear until section 4, and even there, it seems to me more like an application. This is not necessarily a drawback, since it probably shows that the method is quite general. However, I am wondering whether the connection with PDEs should be strengthened. (2) Furthermore, the main contributions are numerical but the only baseline comes from a Kalman filter approach, which is suitable for linear systems but I am not sure if it is suitable for PDEs. I would recommend discussing this choice of baseline and, if possible, to provide other baselines.


**Summary Of The Paper:**

The paper proposes a method to infer the state of a dynamical system based on sparse measurements. More specifically, the authors consider a situation in which there is a dynamical system whose state evolves but we cannot observe it directly. Instead we observe only a low dimensional approximation.. Here, the authors propose a method which combines a reduced order model (ROM) approach with reinforcement learning (RL). Then they provide examples of applications to partial differential equations (PDEs), with numerical results.

**Summary Of The Review:**

The paper is clearly written and the contributions are clearly presented. I think that addressing the two questions mentioned above would help strengthen the paper.

---

> ### Author Response · Authors · 2022-11-17
> **Response to Reviewer TCZP**
>
> We thank the reviewer for finding our paper clearly written and the ideas well presented. Please find below our answers to the reviewer's questions, which helped strengthen the paper. We would also be grateful if the reviewer could mention whether they desire other aspects of the paper to be improved in order to warrant a higher score.
>
> > (1) It seems that the framework and the model purely deal with a discrete time and finite dimensional system. The connection with PDEs (mentioned in the title) is not clear until section 4, and even there, it seems to me more like an application. This is not necessarily a drawback, since it probably shows that the method is quite general. However, I am wondering whether the connection with PDEs should be strengthened.
>
> > To what extent is the proposed method specific to PDEs? It seems that once the PDE is discretized (which is done separately from the proposed method), it is a quite generic method that is developed and applied. In that case, it might be interesting to adjust accordingly the presentation of the paper.
>
> The reviewer is correct that the proposed method can be applied to any high-dimensional system beyond discretized PDEs. Our work is mainly motivated by engineering applications involving PDEs such as active flow control, which is why we introduce the method in the context of PDEs in the title, the abstract and the introduction. We have nevertheless clarified in section 2.1 of the revised version that our work is applicable to any high-dimensional nonlinear system.
>
> > (2) Furthermore, the main contributions are numerical but the only baseline comes from a Kalman filter approach, which is suitable for linear systems but I am not sure if it is suitable for PDEs. I would recommend discussing this choice of baseline and, if possible, to provide other baselines.
>
> > Are there other (better) baselines than a simple Kalman filter? Or could you please provide more details on why this is a good baseline?
>
> We thank the reviewer for raising this excellent question. We have added a paragraph at the beginning of Section 4 discussing different possible baselines and explaining the choice of the Kalman filter as the best baseline in our context.
>
> The ensemble Kalman filter and 4D-Var are two estimation techniques for high-dimensional systems such as those governed by PDEs. Although they are commonly employed for data assimilation in numerical weather prediction, they require large computational resources since they involve repeated solutions of the high-dimensional dynamics model in eq (1). Thus, they are not applicable in the context of embedded control systems, whose limited resources call for an inexpensive model such as the ROM in eq (2). Since the ROM that we consider has linear dynamics, extensions of the Kalman filter for nonlinear dynamics such as the extended or unscented Kalman filters are not relevant, and the linear Kalman filter remains the best choice of baseline.

---

> ### Author Response · Authors · 2022-12-13
> **Follow-up with Reviewer TCZP**
>
> We once again thank the reviewer for their positive comments and relevant questions that lead to several clarifications. We would be grateful if the reviewer could let us know if their initial concerns have been addressed, and we would appreciate any further questions or comments.

---

### Official Review · Reviewer_jaEk · 2022-10-26

**Confidence:** 3
**Correctness:** 3
**Technical Novelty And Significance:** 2
**Empirical Novelty And Significance:** 3
**Recommendation:** 6

**Clarity, Quality, Novelty And Reproducibility:**

The paper is clear in general. RL formulation of estimator is not novel, but its combination with ROM is novel.

**Strength And Weaknesses:**

Strength
- Interesting and important problem
- The general solution approach, using RL for estimation, is definitely exciting

Weakness:
- No discussion on the meaningfulness of the designed estimator. What does the cost in the optimal control represent? Why does minimizing it lead to a good estimator?
- The comparison with KF-ROE can me made more in depth. Are the control laws learned through the RL approach nonlinear? What is the tuning process for KF?
- the inability to consider estimating systems with inputs.
- the discussion about stationary Markov decision problem and the remark about the problem being actually POMDP and non-Markov is a bit confusing.



**Summary Of The Paper:**

The objective of the paper is to construct an estimator for the state of a high-dimensional nonlinear dynamical system given partial observations of the state. The objective is motivated by applications in fluid mechanics or turbulent flows where the state of the system is large obtained by discretizing a PDE. The proposed approach has two main steps:

(1) construction of a reduced order model. In particular, the paper proposes the dynamic mode decomposition method which only requires a single trajectory of the nonlinear dynamics.

(2) formulating the problem of finding the estimator as a MDP problem and application of RL techniques to solve it. In particular, the estimator is modeled as a dynamical system driven by a stochastic control policy that depends on the current value of the estimate and the value of observation. The objective function is modeled with running cost equal to the error in estimating the state and a quadratic penalty on the control. Then, the optimal control policy is learned using a policy gradient method by sampling trajectories from the system.

The proposed approach is evaluated on a benchmark example that involves Burger's equation and compared with Kalman filter applied on the reduced order system.

**Summary Of The Review:**

The paper discusses a very interesting problem and follows a general exciting approach. It should give more motivation for the cost function, and also more in depth comparison to KF, like comparing the learned policy to the best linear policy. Basically, it is interesting to see what would happened if the control action is restricted to be linear and what is the gain from nonlinear parametrization.

---

> ### Author Response · Authors · 2022-11-18
> **Response to Reviewer jaEk (part 1)**
>
> We thank the reviewer for finding our problem interesting and our approach exciting, and for providing interesting suggestions which have helped us improve the quality of the submission. Please find below our answers to the reviewer's comments. We would also be grateful if the reviewer could mention whether they desire other aspects of the paper to be improved in order to warrant a higher score.
>
> > No discussion on the meaningfulness of the designed estimator. What does the cost in the optimal control represent? Why does minimizing it lead to a good estimator?
>
> We agree that such discussion was missing in our original submission. In the revised paper, we clarify the meaningfulness of the proposed estimator by presenting in Appendix B an interpretation of our estimator equations and cost function from the perspective of Bayesian inference. Furthermore, we now state explicitly in eq (5) the optimization problem solved by RL to find the policy parameters. We also note that designing state estimators by correcting the dynamics model with a measurement-dependent term is a standard approach in control theory, where they are called state observers [1]
>
> The Bayesian interpretation demystifies the cost function and shows why its minimization leads to a good estimator. The key is to relate our estimate $\hat{\mathbf{x}}\_k$ at time $k$ to the mean of the posterior pdf $p(\mathbf{x}\_k|\mathbf{y}\_{1:k})$ of the true state given the history of measurements $\mathbf{y}\_{1:k} = \{\mathbf{y}\_1,\dots,\mathbf{y}\_k\}$. This is commonly referred to as the minimum mean square error (MMSE) estimator, which is known to be equivalent to minimizing the variance of the posterior estimate [2,3]. In other words, $\hat{\mathbf{x}}\_k = \mathbb{E}[\mathbf{x}\_k|\mathbf{y}\_{1:k}]$ also minimizes $\mathbb{E}[||\mathbf{x}\_k-\hat{\mathbf{x}}\_k||^2]$ conditioned on $\mathbf{y}\_{1:k}$. By expressing $\hat{\mathbf{x}}\_k = \hat{\mathbf{x}}\_k^- + \boldsymbol{\mu}\_{\boldsymbol{\theta}} (\hat{\mathbf{x}}\_k^-,\mathbf{y}\_k)$, where the prior $\hat{\mathbf{x}}\_k^-$ is obtained by propagating the posterior $\hat{\mathbf{x}}\_{k-1}$ at previous time $k-1$ through the system dynamics, the problem is now to find the parameters $\boldsymbol{\theta}$ of the correction function $\boldsymbol{\mu}\_{\boldsymbol{\theta}}$ so that the posterior variance is minimized. Owing to the stationarity of $\boldsymbol{\mu}\_{\boldsymbol{\theta}}$, we can extend this minimization problem to all time steps, resulting in the minimization problem that we solve with RL in eq (5). This derivation reveals that the cost function derives from the variance minimization principle underlying the definition of the MMSE estimator, which means that its minimization should lead to a good estimator.
>
> [1] Gildas Besançon. Nonlinear observers and applications, volume 363. Springer, 2007.
>
> [2] Simo Sarkka. Bayesian filtering and smoothing. Cambridge university press, 2013.
>
> [3] Steven M Kay. Fundamentals of statistical signal processing: estimation theory. Prentice-Hall, Inc., 1993.
>
> > The comparison with KF-ROE can me made more in depth. Are the control laws learned through the RL approach nonlinear? What is the tuning process for KF?
>
> Following the reviewer's suggestion, we have included in Appendix I of the revised version an evaluation of the nonlinearity of the policy learned through the RL procedure. We therefore quantify the nonlinearity of the policy by evaluating the Jacobian of the mean function $\boldsymbol{\mu}\_{\boldsymbol{\theta}'}(\mathbf{y}\_k,\hat{\mathbf{x}}\_{k-1})$ with respect to its two inputs $\mathbf{y}\_k$ and $\hat{\mathbf{x}}\_{k-1}$. The Jacobian components of a linear policy will be independent of the input values, while the Jacobian components of a nonlinear policy will change with the input values. We show that the distribution of the Frobenius norm of the Jacobian of the trained policies obtained in the Burgers and Navier-Stokes examples has a wide spread, which demonstrates that the mean function $\boldsymbol{\mu}\_{\boldsymbol{\theta}'}$ trained by the RL process is highly nonlinear, as opposed to the linear correction term in the KF-ROE.
>
> We tuned the Kalman filter by following a standard procedure [1]. Since the covariances $\mathbf{Q}_k$ and $\mathbf{R}_k$ of the process noise $\mathbf{w}_k$ and observation noise $\mathbf{v}_k$ are not usually known beforehand, they can be considered as tuning parameters. We therefore assumed that $\mathbf{Q}_k = \beta_Q \mathbf{I}$ and $\mathbf{R}_k = \beta_R \mathbf{I}$, and we did a line search in $[10^{-5}, 10^{10}]$ to find the values of $\beta_Q$ and $\beta_R$ that produced the best estimation performance for the KF-ROE. This resulted in the values $\beta_Q = 10^3$ for Burgers and $\beta_Q = 10^9$ for Navier-Stokes; and $\beta_R = 1$ in both cases. We have updated the submission to provide more details on the tuning process in Appendix C.

---

> ### Author Response · Authors · 2022-11-18
> **Response to Reviewer jaEk (part 2)**
>
> We thank the reviewer for finding our problem interesting and our approach exciting, and for providing interesting suggestions which have helped us improve the quality of the submission. Please find below our answers to the reviewer's comments.
>
> > the inability to consider estimating systems with inputs.
>
> This is indeed a limitation of the current work, as we have stated in section 2.1. Generalizing our approach to systems with exogenous control inputs will be studied in the future.
>
> > the discussion about stationary Markov decision problem and the remark about the problem being actually POMDP and non-Markov is a bit confusing.
>
> We believe that the reviewer is referring to two separate concepts, whose difference is subtle. We have updated the wording in the revised version to try to clarify the confusion.
>
> A stationary MDP means that the transition probabilities and reward function are independent of the time index $k$ [4]. In our work, we first formulate a stationary MDP to solve the optimization problem in eq (5) defining the control policy. This requires absorbing the target state $\mathbf{z}_k$ into the state of the MDP so that the reward function can be independent of time.
>
> But this stationary MDP is in fact a POMDP, for which the state transition function, when expressed in terms of the partial observation, does not satisfy the Markov property of independence on past observations and actions. A globally optimal policy would therefore need to be conditioned on the history of past observations and actions [5]. In practice, we relax the problem and find a locally optimum policy that simply depends on the current observation, as is common practice in the literature [6].
>
> [4] Erwan Lecarpentier and Emmanuel Rachelson. Non-stationary markov decision processes, a worst-case approach using model-based reinforcement learning. Advances in neural information processing systems, 32, 2019.
>
> [5] Leslie Pack Kaelbling, Michael L Littman, and Anthony R Cassandra. Planning and acting in partially observable stochastic domains. Artificial intelligence, 101(1-2):99–134, 1998.
>
> [6] Richard S Sutton and Andrew G Barto. Reinforcement learning: An introduction. MIT press, 2018.
>
> > The paper [...] should give [...] in depth comparison to KF, like comparing the learned policy to the best linear policy. Basically, it is interesting to see what would happened if the control action is restricted to be linear and what is the gain from nonlinear parametrization.
>
> The best linear policy is given by the KF-ROE, since the Kalman filter is the best possible linear estimator [2,3]. Therefore, our comparison of the RL-ROE and KF-ROE in Sections 4.1 and 4.2 demonstrate what is the gain from the nonlinear parameterization. The conclusion from our numerical experiments in challenging settings involving regime bifurcations is that the nonlinearity of the policy yields a large advantage in the sparse limit of very few sensors; allowing the RL-ROE to produce good estimates while the performance of the KF-ROE collapses. On the other hand, in the limit of many sensors, the KF-ROE becomes better than the RL-ROE since the Kalman gain is given by an analytical solution and, therefore, will not converge to a local minimum as happens with the RL-ROE.
>
> > The proposed approach is evaluated on a benchmark example that involves Burger's equation and compared with Kalman filter applied on the reduced order system.
>
> We hope that the reviewer also noticed our example involving the Navier-Stokes equation for the flow past a cylinder. This example is particularly challenging since it involves a bifurcation of the behavior of the system from steady to time-periodic flow as the Reynolds number increases. Even then, the nonlinearity of the policy of the RL-ROE allows it to maintain consistent estimation performance throughout the range of Reynolds numbers, even when the number of sensors becomes very small. In contrast, the linear KF-ROE requires many more sensors to give good estimation accuracy.

---

> ### Author Response · Authors · 2022-12-13
> **Follow-up with Reviewer jaEk**
>
> We once again thank the reviewer for their enthusiastic comments and interesting suggestions that lead to several improvements. We would be grateful if the reviewer could let us know if their initial concerns have been addressed, and we would appreciate any further questions or comments.

---

### Decision · Program_Chairs · 2023-01-20

**Decision:**

Reject

**Justification For Why Not Higher Score:**

See meta-review.

**Justification For Why Not Lower Score:**

n/a

**Metareview: Summary, Strengths And Weaknesses:**

Given a partial state observation in a space-time discretization of the PDE, the paper estimates that PDE.

The paper is interesting and is technically sound.  At the same time it is not very strong, having seemingly ad-hoc derivations.  A theoretical framework based on Bayesian inference would perhaps be more solid.  It was added ex post but more as an add-on, seemingly half-random, than as the theoretical basis of the paper.

All in all, the paper is not strong and convincing enough to merit acceptance at ICLR, but can be rewritten with reasonable effort and submitted to a future venue.

**Summary Of Ac-Reviewer Meeting:**

summarised above